# A Survey on Anti-Spoofing Methods for Facial Recognition with RGB Cameras of Generic Consumer Devices

**DOI:** 10.3390/jimaging6120139

**Published:** 2020-12-15

**Authors:** Zuheng Ming, Muriel Visani, Muhammad Muzzamil Luqman, Jean-Christophe Burie

**Affiliations:** 1L3i Laboratory, La Rochelle University, 17042 La Rochelle, France; zuheng.ming@univ-lr.fr (Z.M.); muhammad_muzzamil.luqman@univ-lr.fr (M.M.L.); jcburie@univ-lr.fr (J.-C.B.); 2School of Information & Communication Technology, Hanoi University of Science and Technology, Hanoi 100000, Vietnam

**Keywords:** biometrics, facial recognition, facial anti-spoofing, facial Presentation Attack Detection (PAD), RGB camera-based anti-spoofing methods, deep learning, survey, computer vision, pattern recognition

## Abstract

The widespread deployment of facial recognition-based biometric systems has made facial presentation attack detection (face anti-spoofing) an increasingly critical issue. This survey thoroughly investigates facial Presentation Attack Detection (PAD) methods that only require RGB cameras of generic consumer devices over the past two decades. We present an attack scenario-oriented typology of the existing facial PAD methods, and we provide a review of over 50 of the most influenced facial PAD methods over the past two decades till today and their related issues. We adopt a comprehensive presentation of the reviewed facial PAD methods following the proposed typology and in chronological order. By doing so, we depict the main challenges, evolutions and current trends in the field of facial PAD and provide insights on its future research. From an experimental point of view, this survey paper provides a summarized overview of the available public databases and an extensive comparison of the results reported in PAD-reviewed papers.

## 1. Introduction

### 1.1. Background

In the past two decades, the advancement of technology in electronics and computer science has provided access to top-level technology devices at affordable prices to an important proportion of the world population. Various biometric systems have been widely deployed in real-life applications, such as online payment and e-commerce security, smartphone-based authentication, secured access control, biometric passport and border checks. Facial recognition is among the most studied biometric technologies since the 90s [1], mainly for its numerous advantages compared to other biometrics. Indeed, faces are highly distinctive among individuals and facial recognition can be implemented even in nonintrusive acquisition scenarios or from a distance.

Recently, deep learning has dramatically improved the state-of-the-art performance of many computer vision tasks, such as image classification and object recognition [2,3,4]. With these significant progresses, facial recognition has also made great breakthroughs such as the success of DeepFace [5], DeepIDs [6], VGG Face [7], FaceNet [8], SphereFace [9] and ArcFace [10]. One of these spectacular breakthroughs occurred between 2014 and 2015, when multiple groups [5,8,11] approached and then surpassed human-level recognition accuracy on very challenging face benchmarks, such as Labeled Faces in the Wild (LFW) [12] or YouTube Faces (YTF) [13]. Thanks to their convenience, excellent performances and great security levels, facial recognition systems are among the most widespread biometric systems in the market compared to other biometrics such as iris and fingerprint recognition [14].

However, given facial authentication systems’ popularity, they became primary targets of Presentation Attacks (PAs) [15]. PAs are performed by malicious or ill-intentioned users who either aim at impersonating someone else’s identity (impersonation attack) or at avoiding being recognized by the system (obfuscation attack). However, compared to face recognition performances, the vulnerabilities of facial authentication systems to PAs have been much less studied.

The main objective of this paper is to present a detailed review of face PAD methods that are crucial for assessing the vulnerability/robustness of current facial recognition-based systems towards ill-intentioned users. Given the prevalence of biometric applications based on facial authentication, such as online payment, it is crucial to protect genuine users against impersonation attacks in real-life scenarios. In this survey paper, we will focus more on impersonation detection. However, at the end of the paper, we will discuss obfuscation detection as well.

The next section provides a categorization of face PAs. Based on this categorization, we will present later in this paper a typology of existing facial PAD methods and then a comprehensive review of such methods, with an extensive comparison of these methods by considering the results reported in the reviewed works.

### 1.2. Categories of Facial Presentation Attacks

One can consider that there are basically two types of Presentation Attacks (PAs).

First, with the advent of internet and social medias where more and more people share photos or videos of their faces, such documents can be used by impostors to try and fool facial authentication systems for impersonation purposes. Such attacks are also called *impersonation* (*spoofing*) attacks.

Second, another (less studied) type of presentation attack is called an *obfuscation* attacks, where a person uses tricks to avoid being recognized by the system (but not necessarily by impersonating a legitimate user’s identity).

In short, while impersonation (spoofing) attacks are generally performed by impostors who are willing to impersonate a legitimate user, obfuscation attacks aim at ensuring that the user remains under the radar of the facial recognition system. Despite their totally different objectives, both types of attacks are listed in the ISO standard [16] dedicated to biometric PAD.

In this survey paper, we focus on impersonation (spoofing) attacks, where the impostor might either use directly biometric data from a legitimate user to mount an attack or to create Presentation Attack Instruments (PAIs, usually spoofs or fakes) that will be used for attacking the face recognition system.

Common PAs/PAIs can generally be categorized as photo attacks, video replay attacks and 3D mask attacks (see Figure 1 for their categorization and Figure 2 for illustrations), whereas obfuscation attacks generally rely on tricks to hide the user’s real identity, such as facial makeup, plastic surgery or face region occlusion.

Photo attacks (sometimes also called print attacks in the literature) and video replay attacks are the most common attacks due to the ever-increasing flow of face images available on the internet and the prevalence of low-cost but high-resolution digital devices. Impostors can simply collect and reuse face samples of genuine users. Photo attacks are carried out by presenting to the facial authentication system a picture of a genuine user. Several strategies are usually used by the impostors. Printed photo attacks (see Figure 2a) consist in presenting a picture printed on a paper (e.g., A3/A4 paper, copper paper or professional photographic paper). On the other hand, in photo display attacks, the picture is displayed on the screen of a digital device such as a smartphone, a tablet or a laptop and then presented to the system. Moreover, as illustrated in Figure 2b, printed photos can be warped (along a vertical and/or horizontal axis) to give some depth to the photo (this strategy is called a warped photo attack). Cut photo attacks consist in using the picture as a photo mask where the mouth, eyes and/or nose regions have been cut out to introduce some liveness cues from the impostor’s face behind the photo, such as eye blinking or mouth movement (see Figure 2c).

Compared to static photo attacks, video replay attacks (see Figure 2d) are more sophisticated, as they introduce intrinsic dynamic information such as eye blinking, mouth movements and changes in facial expressions in order to mimic liveness [21].

Contrary to photo attacks or video replay attacks (that are generally 2D planar attacks, except for warped photo attacks), 3D mask attacks reconstruct 3D face artifacts. One can distinguish between low-quality 3D masks (e.g., crafted from a printed photo as illustrated in Figure 2e) and high-quality 3D masks (e.g., made out of silicone, see Figure 2f). The high realism of the “face-like” 3D structure and the vivid imitation of human skin texture in high-quality 3D masks makes it more challenging to detect 3D mask spoofing by traditional PAD methods (i.e., methods conceived to detect photo or video replay attacks [22,23]). Nowadays, manufacturing a high-quality 3D mask is still expensive [24] and complex and relies on complete 3D acquisition, generally requiring the user’s cooperation [25]. Thus, 3D mask attacks are still far less frequent than photo or video replay attacks. However, with the popularization of 3D acquisition sensors, 3D mask attacks are expected to become more and more frequent in the coming years.

PAD methods for previously unseen attacks (unknown attacks) will be reviewed in Section 2.6, “New trends”, as most of them are still under development and rely on recent approaches such as zero/few-shot learning.

Obfuscation attacks, in which the objective is quite different from impersonation attacks (as the aim for the attacker is to remain unrecognized by the system), generally rely on facial makeup, plastic surgery or face region occlusion (e.g., using accessories such as scarves or sunglasses). However, in some cases, obfuscation attacks can also rely on the use of another person’s biometric data. It fundamentally differs from usual spoofing attacks in its primary objective. However, in some cases, the PAIs for obfuscation attacks can be similar to the ones used for impersonation attacks, e.g., the face mask of another person. While most of the PAD methods reviewed in this paper are usual anti-spoofing methods (for detecting impersonation attacks), obfuscation methods are specifically discussed in Section 5, “Discussion”.

The objective of this paper is to give a review of the impersonation PAD (anti-spoofing) methods that do not require any specific hardware. In other words, we focus on methods that can be implemented with only RGB cameras from Generic Consumer Devices (GCDs). This obviously raises some difficulties and limitations, e.g., when it comes to distinguishing between 2D planar surfaces (photos and screens) and 3D facial surfaces. In the next section, we discuss the motivation for reviewing facial anti-spoofing methods using only GCDs.

### 1.3. Facial PAD Methods with Generic Consumer Devices (GCD)

To the best of our knowledge, there is still no agreed-upon PAD method that can tackle all types of attacks. Given the variety of possible PAs, many facial PAD approaches have been proposed in the past two decades. From a very general perspective, one can distinguish between the methods for facial PAD based on specific hardware/sensors and the approaches using only RGB cameras from GCDs.

Facial PAD methods using specific hardware may rely on structured-light 3D sensors, Time of Flight (ToF) sensors, Near-infrared (NIR) sensors, thermal sensors, etc. In general, such specific sensors considerably facilitate facial PAD. For instance, 3D sensors can discriminate between the 3D face and 2D planar attacks by detecting depth maps [26], while NIR sensors can easily detect video replay attacks (as electronic displays appear almost uniformly dark under NIR illumination) [27,28,29,30], and thermal sensors can detect the characteristic temperature distribution for living faces [31]. Even though such approaches tend to achieve higher performance, they are not yet broadly available to the general public. Indeed, such sensors are still expensive and rarely embedded on ordinary GCDs, with the exception of some costly devices. Therefore, the use of such specific sensors is limited to some applicative scenarios, such as physical access control to protected premises.

However, for most applicative scenarios, the user needs to be authenticated using their own device. In such scenarios, PAD methods that rely on specific hardware are therefore not usable. Thus, researchers and developers widely opt for methods based on RGB cameras that are embedded in most electronic GCDs (such as smartphones, tablets or laptops) [32,33,34,35,36,37].

This is the main reason why, in this work, we focus on the facial PAD approaches that do not require any specific hardware. More precisely, we present a comprehensive review of the research work in facial anti-spoofing methods for facial recognition systems using only the RGB cameras of GCDs. The major contributions of this paper are listed in the section below.

### 1.4. Main Contributions of This Paper

The major contributions of this survey paper are the following:We propose a typology of existing facial PAD methods based on the type of PAs they aim to detect and some specificities of the applicative scenario.We provide a comprehensive review of over 50 recent facial PAD methods that only require (as input) images captured by RGB cameras embedded in most GCDs.We provide a summarized overview of the available public databases for both 2D attacks and 3D mask attacks, which are of vital importance for both model training and testing.We report extensively the results detailed in the reviewed works and quantitatively compare the different PAD methods under uniform benchmarks, metrics and protocols.We discuss some less-studied topics in the field of facial PAD, such as unknown PAs and obfuscation attacks, and we provide some insights for future work.

### 1.5. Structure of This Paper

The remainder of this paper is structured as follows. In Section 2, we propose a typology for facial PAD methods based on RGB cameras from GCDs and review the most representative/recent approaches for each category. In Section 3, we present a summarized overview of the most used/interesting datasets together with their main advantages and limitations. Then, Section 4 presents a comparative evaluation of the reviewed PAD methods. Section 5 provides a discussion about current trends and some insights for future directions of research. Finally, we draw the conclusions in Section 6.

## 2. Overview of Facial PAD Methods Using Only RGB Cameras from GCDs

### 2.1. Typology of Facial PAD Methods

A variety of different typologies could be found in the literature. For instance, Chingovska et al. [37] proposed to group the facial PAD methods into three categories: motion-based, texture-based and image-quality based methods, while Costa-Pazo et al. [38] considered image quality-based facial PAD methods as a subclass of texture-based methods. Ramachandra and Busch [39] classified facial PAD methods into two more general categories: hardware-based and software-based methods. The different approaches are then hierarchically classified into subclasses of these two broad categories. Hernandez-Ortega et al. [40] divided the PAD methods as static or dynamic methods, depending on whether they take into account temporal information. Recently, Bhattacharjee et al. [41] considered the PAD methods as approaches based on visible light and extended-range imagery. The visible light here refers to the range of the electromagnetic spectrum which is typically perceptible by the human visual system such as by cameras from GCDs. Instead, in extended-range imagery, the subject is illuminated under a chosen wavelength band using NIR and SWIR-based cameras with appropriate filters to be able to capture data only in the chosen wavelength band.

Based on the type of attacks presented in Section 1.2 and inspired by [39,41], we categorize facail PAD methods into two broad categories: RGB camera-based PAD methods and PAD methods using specific hardware. As stated earlier, in this paper, we focus on the facial PAD approaches that use only RGB cameras embedded in most GCDs (smartphone, tablet, laptop, etc.). Inside this broad category, we distinguish between the following five different classes:liveness cue-based methods;texture cue-based methods;3D geometric cue-based methods;multiple cues-based methods; andmethods using new trends.

As detailed in Figure 3, each of these five categories is then divided into several subclasses, depending on the applicative scenario or on the type of features/methods used. For each category/subcategory of PAD methods, Table 1 shows the type(s) of PAs that it aims to detect, whereas Figure 3 lists all the facial PAD methods that will be discussed in the remainder of this Section (over 50 methods in total).

From a very general standpoint, we state the following:Liveness cue-based methods aim to detect liveness cues in facial presentation or PAI. The most widely used liveness cues so far are motion (head movements, facial expressions, etc.) and micro-intensity changes corresponding to blood pulse. Thus, liveness cue-based methods can be classified into the following two subcategories:Motion cue-based methods employ motion cues in video clips to discriminate between genuine (alive) faces and static photo attacks. Such methods can effectively in detecting static photo attacks but not video replay with motion/liveness cues and 3D mask attacks;Remote PhotoPlethysmoGraphy (rPPG) is the most widely used technique for measuring facial micro-intensity changes corresponding to blood pulse. rPPG cue-based methods can detect photo and 3D mask attacks, as these PAIs do not show the periodic intensity changes that are characteristic of facial skin. They can also detect “low-quality” video replay attacks that are not able to display those subtle changes (due to the capture conditions and/or PAI characteristics). However, “high-quality” video replay attacks (displaying the dynamic changes of the genuine face’s skin) cannot be detected by rPPG cue-based methods.Texture cue-based methods use static or dynamic texture cues to detect facial PAs by analyzing the micro-texture of the surface presented to the camera. Static texture cues are generally spatial texture features that can be extracted from a single image. In contrast, dynamic texture cues usually consist of spatiotemporal texture features, extracted from an image sequence. Texture cue-based facial PAD methods can detect all types of PAs. However, they might be fooled by “high-quality” 3D masks (masks with a surface texture that mimics good facial texture);Three-dimensional geometric cue-based methods use 3D geometrical features, generally based on the 3D structure or depth information/map of the user’s face or PAIs. Three-dimensional geometric cue-based PAD methods can detect planar photo and video replay attacks but not (in general) 3D mask attacks;Multiple cues-based methods consider different cues (e.g., motion features with texture features) to detect a wider variety of face PAs;Methods using new trends do not necessarily aim to detect specific types of PAs, but their common trait is that they rely on cutting-edge machine learning technology, such as Neural Architecture Search (NAS), zero-shot learning, domain adaption, etc.

In the remainder of this section, we present a detailed review of the over fifty recent PAD methods that are listed in Figure 3, structured using the above typology and in chronological order inside each category/subcategory. In each category, we elaborate both the “conventional” methods that have influenced facial PAD the most and their current evolutions in the deep learning era.

### 2.2. Liveness Cue-Based Methods

Liveness cue-based methods are the first attempt for facial PAD. Liveness cue-based methods aim to detect any dynamic physiological sign of life, such as eye blinking, mouth movement, facial expression changes and pulse beat. They can be categorized as motion-based methods (to detect the eye blinking, mouth movement and facial expression changes) and rPPG-based methods (to detect the pulse beat).

#### 2.2.1. Motion-Based Methods

By detecting movements of the face/facial features, conventional motion-based methods can effectively detect static presentation attacks, such as most photo attacks (without dynamic information). However, they are generally not effective against video replay attacks that display liveness information such as eye blinking, head movements, facial expression changes, etc.

This is why interactive motion-based methods were later introduced, where the user is required to complete a specific (sequence of) movement(s) such as head rotation/tilting, mouth opening, etc. The latter methods are more effective for detecting video replay attacks, but they are intrusive for the user, unlike traditional methods that do not require the user’s collaboration and are therefore nonintrusive.

The rest of this section is structured around two subcategories: (a) nonintrusive motion-based methods, that are more user-friendly and easier to implement, and (b) intrusive/interactive motion-based methods, that are more robust and can detect both static and dynamic PAs.

##### (a) Nonintrusive motion-based methods

Nonintrusive motion-based PAD methods aim to detect intrinsic liveness based on movement (head movement, eye blinking, facial expression changes, etc.).

In 2004, Li et al. [44] first used frequency-based features to detect photo attacks. More specifically, they proposed the Frequency Dynamic Descriptor (FDD), based on frequency components’ energy, to estimate temporal changes due to movements. By setting an FDD threshold, genuine (alive) faces can be distinguished from photo PAs, even for relatively high-resolution photo attacks. This method is easier to implement and is less computationally expensive when compared to the previously proposed motion-based methods that used 3D depth maps to estimate head motions [89,90]. However, its main limitation is that it relies on the assumption that the illumination is invariant during video capture, which cannot always be satisfied in a real-life scenario. This can lead to the presence of a possibly large quantity of “noise” (coming from illumination variations) in the frequency component’s variations, and the method is not conceived to deal with such noise.

Unlike the approach introduced in [44], the method proposed in 2005 by Kollreider et al. [45,46] works directly in the RGB representation space (and not in the frequency domain). More precisely, the authors try to detect the differences in motion patterns between 2D planar photographs and genuine (3D) faces using optical flows. The idea is the following: when a head has a small rotation (which is natural and unintentional), for a real face, the face parts that near the camera (“inner parts”, e.g., nose) move differently from the parts further away from the camera (“outer parts”, e.g., ears). In contrast, a translated photograph generates constant motion among all face regions [45].

More precisely, the authors proposed Optical Flow Lines (OFL), inspired from [91], to measure face motion in horizontal and vertical directions. As illustrated in Figure 4, the different greyscales obtained in the OFL from a genuine (alive face) with a subtle facial rotation reflect the motion differences in between different facial parts, whereas the OFL of a translated photo shows constant motion.

A liveness score in [0, 1] is then calculated from the OFLs of the different facial regions, where 1 indicates that the movement of the surface presented to the camera is coherent with facial movement and 0 indicates that this movement is not coherent with facial movement. By thresholding this liveness score, the method proposed in [45] can detect printed photo attacks, even if the photo is bent or even warped around a cylinder (as it is still far from the real 3D structure of a face). However, this method fails for most video replay attacks, and it can be disrupted by eyeglasses (because they partly cover outer parts of the face but are close to the camera) [46].

In 2009, Bao et al. [47] also leveraged optical flow to distinguish between 3D faces and planar photo attacks. Let us call *O* the object presented to the system (face or planar photo). By comparing the optical flow field of *O* deduced from its perspective projection to a predefined 2D object’s reference optical flow field, the proposed method can determine if the given object is a really a 3D face or a planar photo.

However, like all other optical flow-based methods, the methods in [45,46,47] are not robust toward background and illumination changes.

In 2007, Pan et al. [42,43] chose to focus on eye blinking in order to distinguish between a face and a facial photo. Eye blinking is a physiological behavior that normally happens 15 to 30 times per minute [92]. Therefore, it is possible for GCD cameras having at least 15 frames per second (fps)—which is almost all GCD cameras—to capture two or more frames per blink [42]. Pan et al. [42,43] proposed to use Conditional Random Fields (CRFs) fed by temporally observed images $x_{i}$ to model eye blinking with its different estimated (hidden) states $y_{i}$: Non-Closed (NC, including opened and half-opened eyes), then Closed (C) and then NC again, as illustrated in Figure 5. The authors showed that their CRF-based model (discriminative model) is more effective than Hidden Markov Model (HMM)-based methods [93] (generative model), as it takes into account longer-range dependencies in the data sequence. They also showed that their model is superior to another discriminative model: Viola and Jones’ Adaboost cascade [94] (as the latter is not conceived for sequential data). This method, like all other methods based on eye blinking, can effectively detect printed photo attacks but not video replay attacks or eye-cut photo attacks that simulate blinking [19].

##### (b) Intrusive motion-based methods

Intrusive methods (also called interactive methods) are usually based on a challenge–response mechanism that requires the users to satisfy the system’s requirements. In this paragraph, we present methods where the challenge is based on some predefined head/facial movement (e.g., blinking the eyes, moving the head in a certain way, adopting a given facial expression or uttering a certain sequence of words).

In 2007, Kollreider et al. [48] first proposed an interactive method that can detect replay attacks as well as photo attacks by reading the presented face’s lips when the user is prompted to utter a randomly determined sequence of digits. Like in their previous work [45] mentioned above, Kollreider et al. [48] used OFL to extract mouth motion. An interesting feature of this method is that it combines facial detection and facial PAD in a holistic system. Thus, the integrated facial detection module can also be used to detect the mouth region, and OFL is used for both facial detection and facial PAD.

Then, a 10-class Support Vector Machine (SVM) [95] is trained from 160-dimensional velocity vectors extracted from the mouth region’s OFL to perform recognition of the 0–9 digits. This method detects effectively printed photo attacks and most video replay attacks. However, it is vulnerable to mouth-cut photo attacks, and even though this topic was not studied yet (at least, to the best of our knowledge), it certainly cannot detect sophisticated DeepFakes [96,97,98], where the impostor can “play” on-demand any digit. Another limitation of this method is that it is based on visual cues only, i.e., it does not consider audio together with images (unlike multi-modal audio-visual methods [99]). This makes it vulnerable to “visual-only” DeepFakes, which are of course easier to obtain than realistic audio-visual DeepFakes (with both the facial features and voice of the impersonated genuine user).

More generally, the emerging technology of DeepFakes [100] is a great challenge for interactive motion-based PAD methods. Indeed, based on deep learning models such as autoencoders and generative adversarial networks [101,102], DeepFakes can superimpose face images of a target person to a video of a source person in order to create a video of the target person doing or saying things that the source person does or says [96,97,98]. Impostors can therefore use DeepFake generation apps like FaceApp [103] or FakeApp [104] to easily create a video replay attack showing a genuine user’s face satisfying the system requirements during the challenge–response authentication. Interactive motion-based methods generally have difficulties detecting DeepFakes-based video replay attacks.

However, recent works show that rPPG-based [105,106] and texture-based methods [107] can be used to detect video attacks generated using DeepFakes. rPPG-based methods are discussed in the next section.

#### 2.2.2. Liveness Detection Based on Remote PhotoPlethysmoGraphy (rPPG)

Unlike head/facial movements that are relatively easy to detect, intensity changes in the facial skin that are characteristic of pulse/heartbeat are imperceptible for most human eyes. To detect these subtle changes automatically, remote PhotoPlethysmoGraphy (rPPG) was proposed [17,33,49]. rPPG can detect blood flow using only RGB images from a distance (in a nonintrusive way) based on the analysis of variations in the absorption and reflection of light passing through human skin. The idea behind rPGG is illustrated in Figure 6.

Since photo-based PAs do not display any periodical variation in the rPPG signal, they can be detected easily by rPPG-based methods. Moreover, as illustrated in Figure 6, most kinds of 3D masks (including high-quality masks, except maybe extremely thin masks) can be detected by rPPG-based methods. However, “high-quality” video replay attacks (with good capture conditions and good-quality PAI) can also display periodic variation of the genuine face’s skin light absorption/reflection. Thus, rPPG-based methods are only capable in detecting low-quality video replay attacks.

The first methods that applied rPPG to facial PAD were published in 2016. Li et al. [49] proposed a simple approach for which the framework is shown in Figure 7. The lower half of the face is detected and extracted as a Region of Interest (RoI). The rPPG signal is composed of the average RGB value of pixels in the ROI for each RGB channel of each video frame. This rPPG signal is then filtered (to remove noise and to extract the normal pulse range) and transformed into a frequency signal by Fast Fourier Transform (FFT). Two frequency features per channel (denoted as $E_{r},E_{g}$ and $E_{b}$ and as $\Gamma_{r},\Gamma_{g}$ and $\Gamma_{b}]$ in Figure 7) are extracted for each color channel based on the Power Spectral Density (PSD). Finally, these (concatenated) feature vectors are fed into an SVM to differentiate genuine facial presentation from PAs.

This rPPG-based method can effectively detect photo-based and 3D mask attacks—even high-quality 3D masks—but not (in general) video replay attacks. Because, on the other hand, texture-based methods (see Section 2.3) can detect video replay attacks but not realistic 3D masks [25,35,55], the authors also proposed a cascade system that uses first their rPPG-based method (to filter photo or 3D mask attacks) and then a texture-based method (to detect video replay attacks).

Also in 2016, Liu et al. [51] proposed another rPPG-based method for detecting 3D mask attacks. Its principle is illustrated in Figure 8. This method has three interesting features compared to the abovementioned approach proposed in [49] the same year. First, rPPG signals were extracted from multiple facial regions instead of just the lower half of the face. Secondly, the correlation of any two local rPPG signals was used as a discriminative feature (assuming they should all be consistent with the heartbeat’s rhythm). Thirdly, a confidence map is learned, o weigh each region’s contribution: robust regions that contain strong heartbeat signals are emphasized, whereas unreliable regions containing less heartbeat signals (or more noise) are weakened.

Finally, the weighted local correlation-based features are fed into an SVM (with Radial Basis Function (RBF) kernel) to detect photo and 3D mask PAs. This approach is more effective than the one proposed by Li et al. in [49].

In 2017, Nowara et al. [50] proposed PPGSecure, a local rPPG-based approach within a framework that is very similar to the one in [49] (see Figure 7). The rPPG signals are extracted from three facial regions (forehead and left/right cheeks) and two background regions (on the left and right sides of the facial region). The use of background regions provides robustness against noise due to illumination fluctuations, as this noise can be subtracted from the facial regions after having been detected in the background regions. Finally, the Fourier spectrum’s magnitudes of the filtered rPPG signals are fed into an SVM or a Random Forest Classifier [108]. The authors showed experimentally the interest of using background regions, and their method obtained better performances than the one in [51] on some dataset.

In 2018, Liu et al. [33] proposed a deep learning-based approach that can learn rPPG signals in a robust way (under different poses, illumination conditions and facial expression). In this approach, rPPG estimations (pseudo-rPPGs) were combined with the estimations of 3D geometric cues in order to tackle not only photo and 3D mask attacks (like all rPPG-based methods) but also video replay attacks. Therefore, this approach is detailed together with other multiple cue-based approaches, in Section 2.5.2, on page 25.

As mentioned in Section 2.2.1, recent studies show that, unlike motion-based PAD methods (e.g., the interactive motion-based PAD method, such as in Kollreider et al. [48], with better generalization based on challenge–response mechanism), rPPG-based PAD methods can be used to detect DeepFake videos.

Indeed, in 2019, Fernandes et al. [105] proposed to use Neural Ordinary Differential Equations (Neural-ODE) [109] for heart rate prediction. The model is trained on the heart rate extracted from the original videos. Then, the trained Neural-ODE is used to predict the heart rate of Deepfake videos generated from these original videos. The authors show that there is a significant difference between the original videos’ heart rates and their predictions in the case of DeepFakes, implicitly showing that their method could discriminate between Deepfakes and genuine videos.

A more sophisticated method was proposed in 2020 by Ciftci et al. [106], in which several biological signals (including the rPPG signal) are fed into a specifically designed Convolutional Neural Network (CNN) to discriminate between genuine videos and DeepFakes. The reported evaluation results are very encouraging.

Either with the motion-based approaches or with the rPPG-based approaches, the liveness cue-based methods need a video clip accumulating enough numbers of video frames to detect dynamic biometric traits such as eye blinking, head movement or intensity changes in the facial skin. Thus, the duration of the videos needed to assess liveness makes liveness cue-based methods less user-friendly and hard to be applied in real-time scenarios.


### 2.3. Texture Cue-Based Methods

Texture feature-based methods are the most widely used for facial PAD so far. Indeed, they have several advantages compared to other kinds of methods. First, they are inherently nonintrusive. Second, they are capable to detect almost any kind of known attacks, e.g., photo-based attacks, video replay attacks and even some 3D mask attacks.

Unlike the liveness cue-based methods that rely on dynamic physiological signs of life, texture cue-based methods explore the texture properties of the object presented to the system (genuine face or PAI). With texture cue-based methods, PAD is usually formalized as a binary classification problem (real face/non-face) and these methods generally rely on a discriminative model.

Texture cue-based methods can be categorized as static texture-based and dynamic texture-based. Static texture-based methods extract spatial or frequential features, generally from a single image. In contrast, dynamic texture-based methods explore spatiotemporal features extracted from video sequences. The next two subsections present the most prominent approaches from these two types.

#### 2.3.1. Static Texture-Based Methods

The first attempt to use static texture clues for facial PAD dates back to 2004 [44]. In this method, the difference of light reflectivity between a genuine (alive) face and its printed photo is analyzed using their frequency representations (and, more specifically, their 2D Fourier spectra). Indeed, as illustrated in Figure 9, the 2D Fourier spectrum of a face picture has much less high-frequency components than the 2D Fourier spectrum of a genuine (alive) facial image.

More specifically, the method relies on High-Frequency Descriptors (HFD), defined as the energy percentage explained by high-frequency components in the 2D Fourier spectrum. Then, printed photo attacks are detected by thresholding the HFD value (attacks being below the threshold). This method works well only for small images with poor resolution. For instance, it is vulnerable to photos of 124 × 84 mm or with 600 dpi resolution.

In 2010, Tan et al. [52] first modeled the respective reflectivities of images of genuine (alive) faces and printed photos using physical models (here Lambertian models) [110], in which latent samples are derived using Difference of Gaussian (DoG) filtering [111]. The idea behind this method is that an image of a printed photo tends to be more distorted than an image of a real face because it has been captured twice (by possibly different sensors) and printed once (see Section 3.1 for more information about the capture process), whereas real faces are only captured once (by the biometric system only). Several classifiers were tested, among which Sparse Nonlinear Logistic Regression (SNLR) and SVMs, with SNLR proving to be slightly more effective.

Since DoG filtering is sensitive to illumination variations and partial occlusion, Peixoto et al. [53] proposed in 2011 to apply Contrast-Limited Adaptive Histogram Equalization (CLAHE) [112] to preprocess all images, showing the superiority of CLAHE to a simple histogram equalization.

Similar to the work from Tan et al. [52] in 2010, Bai et al. [54] also used, in the same year, a physical model to analyze the images’ micro-textures, using Bidirectional Reflectance Distribution Functions (BRDF). The original image’s normalized specular component (called *specular ratio image*) is extracted [113], and then its gradient histogram (called *specular gradient histogram*) is calculated. As shown in Figure 10, the shapes of the specular gradient histograms of a genuine (alive) face and of a printed photo are quite different. To characterize the shape of a specular gradient histogram, a Rayleigh histogram model is fitted on the gradient histogram. Then, its two estimated parameters σ and β are used to feed an SVM. This SVM is trained to discriminate between genuine face images and planar PAs (in particular, printed photos and video replay attacks).

As shown in Figure 10, this method can detect planar attacks just from a small patch of the image. However, specular component extraction requires a highly contrasted image, and therefore, this method is vulnerable towards any kind of blur.

Local Binary Pattern (LBP) [114] is one of the most widely used hand-crafted texture features in face analysis-related problems, such as face recognition [115], face detection [116] and facial expression recognition [117]. Indeed, it has several advantages, including a certain robustness toward illumination variations.

In 2011, Määttä et al. [55] first proposed to apply multi-scale LBP to face PAD. Unlike the previously described static texture-based approaches [52,54], LBP-based methods do not rely on any physical model; they just assume that the differences in surface properties and light reflection between a genuine face and a planar attack can be captured by LBP features.

Figure 11 illustrates this method. Three different LBPs were applied on a normalized 64 × 64 image in [55]: $LBP_{8,2}^{u2}$, a uniform circular LBP extracted from an 8-pixel neighbourhood with a 2-pixel radius; $LBP_{16,2}^{u2}$, a uniform circular LBP extracted from a 16-pixel neighbourhood with a 2-pixel radius; and $LBP_{8,1}^{u2}$, a uniform circular LBP extracted from a 8 pixel neighbourhood with a 1-pixel radius. Finally, a concatenation of all generated histograms formed a 833-bin/dimension histogram. This histogram is then used as a global micro-texture feature and fed to a nonlinear (RBF) SVM classifier for facial PAD.

In 2012, the authors extended their work in [61], adding two more texture features within the same framework: Gabor wavelets [118] (that can describe facial macroscopic information) and Histogram of Oriented Gradients (HOG) [119] (that can capture the face’s edges or gradient structures). Each feature (LBP-based global micro-texture feature, Gabor wavelets and HOG) is transformed into a compact linear representation by using a homogeneous kernel map function [120]. Then, each transformed feature is separately fed into a fast linear SVM. Finally, late fusion between the scores of the three SVM output is applied to generate a final decision. The authors showed the superiority of this approach compared to the method they previously introduced in [55].

In 2013, the same authors continued to extend their work [63], using this time the upper-body region instead of the face region to detect spoofing attacks. As shown in Figure 12, the upper-body region includes more scenic cues of the context, which enables the detection of the boundaries of the PAI (e.g., video screen frame or photograph edge), and, possibly, the impostor’s hand(s) holding the PAI. As a local shape feature, HOG is calculated from the upper-body region to capture the continuous edges of the PAI (see Figure 12d). Then, this HOG feature is fed to a linear SVM for detecting photo or video replay attacks. The upper-body region is detected using the method in [121], that can also be used to filter poor attacks (where the PAI is poorly positioned or with strong discontinuities between the face and shoulder regions), as shown in Figure 12c.

In the same spirit of using the context surrounding a face, Yang et al. [62] proposed (also in 2013) to use a 1.6× enlarged face region, called Holistic-Face (H-Face), to perform PAD. In order to focus on the facial regions that play the most important role in facial PAD, the authors segmented four canonical facial regions: the left eye region, right eye region, nose region and mouth region, as shown in Figure 13. The rest of the face (mainly the facial contour region) and the original enlarged facial images were divided as 2×2 blocks to obtain another eight components. Thus, twelve face components are used in total. Then, different texture features such as LBP [55], HOG [119] and Local Phase Quantization (LPQ) [122] are extracted as low-level features from each component. Instead of directly feeding the low-level features to the classifier, a high-level descriptor is generated based on the low-level features by using spatial pyramids [123] with a 512-word codebook. Then, the high-level descriptors are weighted using average pooling to extract higher-level image representations. Finally, the histogram of these image representations are concatenated into a single feature vector fed into an SVM classifier to detect PAs.

In 2013, Kose et al. [56] first proposed a static texture-based approach to detect 3D mask attacks. Due to the unavailability of public mask attack databases at that time, 3D mask PADs were much less studied than photo or video replay attacks. In this work, the LBP-based method in [55] is directly applied to detect 3D mask attacks by using the texture (original) image or depth image of the 3D mask attacks (from a self-constructed database), as shown in Figure 14. Note that all the texture images and the depth images were obtained by MORPHO (http://www.morpho.com/). This work showed that using the texture (original) image is better than using a depth image for detecting 3D mask attacks with LBP features. The authors also proposed in [57] to improve this method by fusing the LBP features of the texture image and depth image. They showed the superiority of this approach toward the previous method (using only texture images).

Erdogmus et al. [25,58] also proposed in 2013 a method for 3D mask attack detection based on LBP. They used different classifiers, such as Linear Discriminant Analysis (LDA) and SVMs. On the proposed 3D Mask Attack Database (3DMAD), which is also the first public spoofing database for 3D mask attacks, LDA was proved to be best among the tested classifiers.

Galbally et al. [64,65] introduced in 2013 and 2014 new facial PAD methods based on Image Quality Assessment (IQA), assuming that a spoofing image captured in a photo or video replay PA should have a different quality than a real sample, as it was captured twice instead of once for genuine faces (this idea is similar to the underlying idea of the method in [52] presented above). The quality differences concern sharpness, colour and luminance levels, structural distortions, etc. Fourteen and twenty-five image quality measures were adopted in [64,65], respectively, to assess the image quality using scores extracted from single images. Then, the image-quality scores were combined as a single feature vector and fed into an LDA or Quadratic Discriminant Analysis (QDA) classifier to perform facial PAD. The major advantage of the IQA-based methods is that it is not a trait-specific method, i.e., it does not rely on a priori face/body region detection, so this is a “multi-biometric” method that can also be employed for iris or fingerprint-based liveness detection. However, the performance of the proposed IQA-based methods for PAD was limited compared to other texture-based methods, and the method is not conceived to detect 3D mask attacks.

In 2015, Wen et al. [32] also proposed an IQA-based method, using analysis of image distortion, for facial PAD. Unlike the methods from Tan et al. [111] and Bai et al. [54] presented above—methods that work in the RGB space—this method analyzes the image chromaticity and the colour diversity distortion in the HSV (Hue, Saturation and Value) space. Indeed, when the input image resolution is not enough, it is hard to tell the difference between a genuine face and a PA based only on the RGB image (or grey-scale image). The idea here is to detect imperfect/limited colour rendering of a printer or LCD screen. A 121-dimensional image distortion feature (which consists of a three-dimensional specular reflection feature [124], a two-dimensional no-reference blurriness feature [125,126], a 15-dimensional chromatic moment feature [127] and a 101-dimensional colour diversity feature) is fed into two SVMs corresponding respectively to photo attacks and video replay attacks. Finally, a score-level fusion based on the Min Rule [128] gives the final result. Unlike the IQA score-based features used in [65], this feature is face-specific. The proposed method has shown a promising generalization performance when compared with other texture-based PAD methods.

In 2015, Boulkenafet et al. [35,60] also proposed to extract LBP features in HSV or YCbCr colour spaces. Indeed, subtle differences between a genuine face and a PA can be captured by chroma characteristics, such as the Cr channel that is separated from the luminance in the YCbCr colour space (see Figure 15). By simply changing the colour space used, this LBP-based method achieved state-of-the-art performances when compared with some much more complicated PAD methods based on Component Dependent Descriptor (CDD) [62] and even the emerging deep CNNs [34]. This work showed the interest of using diverse colour spaces for facial PAD.

In 2016, Patel et al. [66] first proposed a spoof detection approach on a smartphone. They used a concatenation of multi-scale LBP [55] and image-quality-based colour moment features [32] as a single feature vector fed into an SVM for facial PAD. Like in [63], this work also introduced a strategy to prefilter poor attacks before employing the sophisticated SVM for facial PAD. For this purpose, the authors proposed to detect the bezel of PAI (e.g., the white bezel of photos or the screen’s black bezel along the border) and the Inter-Pupillary Distance (IPD). For bezel detection, if the pixel intensity values remain fairly consistent (over 60 or 50 pixels) on any four sides (top, bottom, left and right sides), the region is considered as belonging to the bezel of a PAI. For IPD detection, if the IPD is too small (i.e., the PAI is too far from the acquisition camera) or too large (i.e., the PAI is too close to the acquisition camera), then the presentation is classified as a PA. The threshold is set to a difference exceeding two times the IPD’s standard deviation observed for genuine faces with a smartphone. This strategy, relying on two simple countermeasures, can efficiently filter almost 95% of the poorest attacks. However, it may generate false rejections (e.g., if the genuine user is wearing a black t-shirt on a dark background).

More recently, deep learning-based methods are used to learn automatically the texture features. Researchers studying these techniques focus on designing an appropriate neural network so as to learn the best texture features rather than to design the texture features themselves (as is the case with most hand-crafted features presented above).

The first attempt to use Convolutional Neural Networks (CNNs) for detecting spoofing attacks was claimed by Yang et al. for their 2014 method [34]. In this method, a one-path AlexNet [2] is used for learning the texture features that best discriminate PAs (see Figure 16). The usual output of AlexNet (a 1000-way softmax) is replaced by an SVM with binary classes. The fully connected bottleneck layer, i.e., fc7, is extracted as the learned texture feature and fed into the binary SVM. Instead of being an end-to-end framework like many CNN frameworks used nowadays, the proposed approach was basically using a quite conventional SVM-based general framework, only replacing the hand-crafted features with the features learned by AlexNet. This method was shown to attain significant improvements when the input image was enlarged by a scale of 1.8 or 2.6. These results are consistent with the previous studies in [62,63] that had already shown that including more context information from the background can help facial PAD. It was the first time that CNNs were proven to be effective for automatically learning texture features for face PAD. This method has surpassed almost all the existing state-of-the-art methods for photo and video replay attacks. It showed the potential of deep CNNs for face PAD. Later, more and more CNN-based methods were explored for facial PAD.

In 2016, Patel et al. [67] first proposed an end-to-end framework based on one-path AlexNet [2], namely CaffeNet, for facial PAD. A two-way softmax replaced the original 1000-way softmax as a binary classifier. Given the small sizes of existing face spoof databases, especially at that time, such deep CNNs were likely to overfit [67] if trained on such datasets. Therefore, the proposed CNN was pretrained on ImageNet [129] and WebFace [130] to provide a reasonable initialization and fine-tuned using the existing face PAD databases. More specifically, two separate CNNs are trained, respectively from aligned face images and enlarged images including some background. Finally, a voting fusion is used to generate a final decision. Just like Yang et al.’s method [34], the proposed CNN-based method has surpassed the state-of-the-art methods based on hand-crafted features for photo and video replay attacks.

Also in 2016, Li et al. [68] proposed to train a deep CNN based on VGG-Face [7] for facial PAD. As in [67], the CNN was pretrained on massive datasets and fine-tuned on the (way smaller) facial spoofing database. Furthermore, the features extracted from the different layers of the CNN were fused to a single feature and fed into an SVM for facial PAD. However, as the dimension of the fused feature is much higher than the number of training samples, this approach is prone to overfitting. Principal component analysis (PCA) and the so-called *part features* are therefore used to reduce the feature dimension. To obtain part features, the mean feature map in a given layer is firstly calculated. Then, the critical positions in the mean feature map are selected, in which the values are higher than 0.9 times the maximum value in the mean feature map. Finally, the values of the critical positions on each feature map are selected to generate the part feature. The concatenation of all part features of all feature maps is used as the global part feature. Then, PCA is applied on the global part feature to further reduce the dimension. Finally, the condensed part feature is fed into an SVM to discriminate between genuine (real) faces and PAs. Benefitting from using a deeper CNN based on VGG-Face, the proposed method has achieved state-of-the-art performances in both intra-dataset and cross-dataset scenarios (see Section 4) for detecting photo and video replay attacks.

In 2018, Jourabloo et al. [69] proposed to estimate the noise of a given spoof face image to detect photo/video replay attacks (the authors also claimed that the proposed method could be applied to detect makeup attacks). In this work, the spoof image was regarded as the summation of the genuine image and image-dependent noise (e.g., blurring, reflection and moiré pattern) introduced when generating the spoof image. Since the noise of a genuine image was assumed as zero in this work, a spoof image can be detected by thresholding the estimated noise. A GAN framework based on CNNs, De-Spoof Net (DS Net), was proposed to estimate the noise. However, as there is no noise ground-truth, instead of assessing the quality of noise estimation, the authors de-noise the spoof images and assess the quality of the recovered (de-noised) image using Discriminative Quality Net (DQ Net) and Visual Quality Net (VQ Net). Besides, by fusing different losses for modelling different noise patterns in DS Net, the proposed method has shown a superior performance compared to other state-of-the-art deep facial PAD methods such as in [33].

In 2019, George et al. [70] proposed Deep Pixelwise Binary Supervision (DeepPixBiS), based on DenseNet [131], for facial PAD. Instead of only using the binary cross-entropy loss of the final output (denoted as Loss 2 in Figure 17) as in [67], DeepPixBiS also uses during training a pixel-wise binary cross-entropy loss based on the last feature map (denoted as Loss 1 in Figure 17). Each pixel in the feature map is annotated as 1 for a genuine face input and as 0 for a spoof face input. In the evaluation/test phase, only the mean value of pixels in the feature map is used as the score for facial PAD. Thanks to the powerful DenseNet and the proposed pixel-wise loss forcing the network to learn the patch-wise feature, DeepPixBiS showed a promising PAD performance for both photo and video replay attacks.

#### 2.3.2. Dynamic Texture-Based Methods

Unlike the static texture-based methods that extract spatial features usually based on a single image, dynamic texture-based methods extract spatiotemporal features using an image sequence.

Pereira et al. [71,72] first proposed in 2012 and 2014 the application of a dynamic texture based on LBP [114] for facial PAD. More precisely, they introduced the LBP from the Three Orthogonal Planes (LBP-TOP) feature [117]. LBP-TOP is a spatiotemporal texture feature extracted from an image sequence considering three orthogonal planes intersecting at the current pixel in the XY direction (as in traditional LBP) and in the XT and YT directions, where T is the time axis of the image sequence, as shown in Figure 18. The sizes of the XT and YT planes depend on the radii in the direction of time axis T, which is indeed the number of frames before or after the central frame in the image sequence. Then, the conventional LBP operation can be applied to each of the three planes. The concatenation of the three LBP features extracted from the three orthogonal planes generates the LBP-TOP feature of the current image. Similar to many static LBP-based facial PAD methods, LBP-TOP is then fed into a classifier such as SVM, LDA or $\chi^{2}$ distance-based classifier to perform facial PAD.

In 2013, the same authors extended their work [73] by proposing two new training strategies to improve generalization. One strategy was to train the model with the combination of multiple datasets. The other was to use a score level fusion-based framework, in which the model was trained on each dataset, and a sum of the normalized score of each trained model was used as the final output. Despite the fact that these two strategies somehow ameliorate generalization, they have obvious drawbacks. First, even a combination of multiple databases cannot deal with new types of attacks that are not included in the current training datasets, so the model has to be retrained when a new attack type is added. Second, the fusion strategy relies on an assumption of statistical independence that is not necessarily verified in practice.

Also in 2013, Bharadwaj et al. [74] proposed to use motion magnification [132] as preprocessing to enhance the intensity value of motion in the video before extracting the texture features. The authors claimed that the motion magnification might enrich the texture of the magnified video. The authors proposed to apply Histogram of Oriented Optical Flows (HOOF) [133] on the enhanced video to conduct facial PAD. HOOF calculates the optical flow between frames at a fixed interval and collects the optical flow orientation angle weighted by its magnitude in a histogram. The histogram is computed from local blocks, and the resulting histograms for each block are concatenated to form a single feature vector as shown in Figure 19. HOOF is much computationally lighter than LBP-TOP. However, the proposed method based on motion magnification needs to accumulate a large number of video frames (>200 frames), which makes it hardly applicable real-time, resulting in solutions that are not very user-friendly.

In 2012 and 2015, Pinto et al. [75,76,77] proposed a PAD method based on the analysis of a video’s Fourier spectrum. Instead of analyzing the Fourier spectra of the original video as in Li et al. [44], the proposed method analyzed the Fourier spectra of the residual noise videos, which only include noise information. The objective is to capture the effect of the noise introduced by the spoofing attack, e.g., the moiré pattern effect shown in Figure 20b,c. In order to obtain a residual noise video, the original video is first submitted to a filtering processing (e.g., Gaussian filter or Median filter). Then, a subtraction is performed between the original and the filtered video, resulting in the noise residual video. Given that the highest responses representing the noise are concentrated on the abscissa and ordinate axes of the logarithm of the Fourier spectrum, visual rhythms [134,135] are constructed to capture temporal information of the spectrum video sampling the central horizontal lines or central vertical lines of each frame and concatenating the sampling lines in a single image, called horizontal or vertical visual rhythm. Then, the grey-level cooccurrence matrices (GLCM) [136], LBP and HOG can be calculated on the visual rhythm as the texture features and can be fed into an SVM or Partial Least Square (PLS). Furthermore, a more sophisticated method, based on the Bag-of-Visual-Word model [137], similar to the Vector Quantization (VQ) [123] used in [62], was also applied to extract the mid-level descriptor base on the low-level features, e.g., LBP and HoG, extracted from the Fourier spectrum.

Also in 2012 and 2015 [75,76], Tirunagari et al. [23] proposed to represent the dynamic characteristics of a video by a single image using Dynamic Mode Decomposition (DMD) [138]. Instead of sampling central lines of each frame in a video spectrum and concatenating them in a single frame (as in visual rhythms), the proposed approach selects the most representative frame in a video generated from the original video by applying DMD in the spatial space. DMD, similarly to Principal Component Analysis (PCA), is based on eigenvalues but, contrary to PCA, it can capture the motion in videos. The LBP feature of the DMD image is then calculated and fed into an SVM for facial PAD.

Xu et al. [78] first proposed to apply deep learning to learn the spatiotemporal features of a video for facial PAD in 2015. More specifically, they proposed an architecture based on Long Short-Term Memory (LSTM) and CNN networks. As shown in Figure 21, several CNN-based branches with only two convolutional layers are used. Each branch is used to extract the spatial texture features of one frame. These frames are sampled from the input video using a certain time step. Then, the LSTM units are connected at the end of each CNN branch to learn the temporal relations between frames. Finally, all the outputs of the LSTM units are connected to a softmax layer that gives the final classification of the input video for facial PAD. Like several researchers before them, the authors also observed that using the scaled image of an original detected face including more background information can help in facial PAD.

In 2019, Yang et al. [79] proposed a Spatiotemporal Anti-Spoofing Network (STASN) to detect photo and video replay PAs. STASN consists of three modules: Temporal Anti-Spoofing Module (TASM), Region Attention Module (RAM) and Spatial Anti-Spoofing Module (SASM). The proposed TASM is composed of CNN and LSTM units to learn the temporal features of the input video. One significant contribution is that, instead of using local regions with predefined locations as in [51,62], STASN uses *K* local regions of the image selected automatically by RAM and TASM based on attention mechanism. These regions are then fed into SASM (i.e., a CNN with *K* branches) for learning spatial texture features. STASN has significantly improved the performance for facial PAD, especially in terms of the generalization capacity shown in cross-database evaluation scenarios (see Section 4).

### 2.4. 3D Geometric Cue-Based Methods

Three-dimensional geometric cue-based PAD methods use 3D geometric features to discriminate between a genuine face with a 3D structure that is characteristic of a face and a 2D planar PA (e.g., a photo or video replay attack). The most widely used 3D geometric cues are the 3D shape reconstructed from the 2D image captured by the RGB camera and the facial depth map, i.e., the distance between the camera and each pixel in the facial region. The two following subsections discuss the approaches based on these two cues, respectively.

#### 2.4.1. 3D Shape-Based Methods

In 2013, Wang et al. [80] proposed a 3D shape-based method to detect photo attacks, in which the 3D facial structure is reconstructed from 2D facial landmarks [139] detected using different viewpoints [54,140]. As shown in Figure 22, the reconstructed 3D structures of a real face and a planar photo are different. In particular, the reconstructed 3D structure from a real face profile preserves its 3D geometric structure. In contrast, the reconstructed structure of a planar photo in profile view is only a line showing the photo’s edge. The concatenation of the 3D coordinates of the reconstructed sparse structure are used as 3D geometric features and fed into an SVM for face PAD. A drawback of this approach is that it requires multiple viewpoints and cannot be used from a single image; using not enough key frames can lead to inaccuracies in 3D structure reconstruction. Moreover, it is susceptible to inaccuracies in the detection of facial landmarks.

#### 2.4.2. Pseudo-Depth Map-Based Methods

The depth map is defined as the distance of the face to the camera. Obviously, when using specific 3D sensors, the depth map can be captured directly. However, in this survey, we focus on approaches that can be applied using GCDs that do not usually embed 3D sensors. However, thanks to significant progress in the computer vision area, especially with the deep learning technology, it is possible to get a good reconstruction/estimation of a depth map from a single RGB image [141,142,143]. Such reconstructions are called *pseudo-depth maps*. Based on the pseudo-depth map of a given image, the different PAD methods can be designed to discriminate between genuine faces and planar PAs.

In 2017, Atoum et al. [36] first proposed a depth-map-based PAD method to detect planar face PADs, e.g., printed photo attack and video replay attacks. The idea is to use the fact that the depth map of an actual face has varying height values in the depth map, whereas planar attacks’ depth maps are constant (see Figure 23) to distinguish between real 3D faces and planar PAs. In this work, an 11-layer fully connected CNN [144] for which the parameters are independent of the size of the input facial images is proposed to estimate the depth map of a given image. The ground truth of depth maps was estimated using a state-of-the-art 3D face model fitting algorithm [142,145,146] for real faces, while it was set to zero for planar PAs, as shown in Figure 23. Finally, the estimated depth maps are fed to a SVM (pretrained using the ground truth) to detect planar face PAs.

In 2018, Wang et al. [81] extended the single frame-based depth-map PAD method in [36] to videos by proposing Face Anti-Spoofing Temporal-Depth networks (FAS-TD). FAS-TD networks are used to capture the motion and depth information of a given video. By integrating Optical Flow guided Feature Block (OFFB) and Convolution Gated Recurrent Unit (ConvGRU) modules to a depth-supervised neural network architecture, the proposed FAS-TD can capture short-term and long-term motion patterns of real faces and planar PAs in videos well. The proposed FAS-TD further improved the performance of the depth-map-based PAD methods using a single frame as in [33,36] and achieved state-of-the-art performances.

Since pseudo-depth map approaches are very effective for detecting planar PAs, pseudo-depth maps are often used in conjunction with other cues in the multiple cues-based PAD method. Also, as pseudo-depth maps are among the most recently introduced cues for facial PAD, they are extensively used in the most recent approaches. These two points are further detailed in the following Section 2.5 and Section 2.6, respectively.

### 2.5. Multiple Cue-Based Methods

Multi-modal systems are intrinsically more difficult to spoof than uni-modal systems [61,147]. Some attempts to counterfeit facial spoofing therefore combine methods based on different modalities, such as visible infrared [22], thermal infrared [148] or 3D [25] signals. However, the fact that such specific hardware is generally unavailable in most GCDs prevents these multi-modal solutions from being integrated into most existing facial recognition systems. In this work, we focus on the multiple cues-based methods that use only images acquired using RGB cameras.

Such multiple cues-based methods combine liveness cues, texture cues and/or 3D geometric cues to address the detection of various types of facial PAs. In general, late fusion is used to merge the scores obtained from the different cues to determine if the input image corresponds to a real face.

#### 2.5.1. Fusion of Liveness Cue and Texture Cues

In this section, the motion cue is used as a liveness cue in conjunction with different texture cues.

In 2017, Pan et al. [83] proposed to jointly use eye-blinking detection and the texture-based scene context matching for facial PAD. The Conditional Random Field (CRF)-based eye-blinking model proposed by the same authors [42] (see page 9) is used to detect eye blinking. Then, a texture-based method is proposed to check the coherence between the background region and the actual background (reference image). The reference image is acquired by taking a picture of the background without the user being present. If the attempt is a real facial presentation, the background region around the face in the reference image and the input image should theoretically be identical. Contrarily, if a video or recaptured photo (printed or displayed on a screen) is presented before the camera, then the background region around the face should be different between the reference image and the input image. To perform a comparison between the input image’s background and the reference image, LBP features are extracted from several fiducial points selected using the DoG function [149] and used to calculate the $\chi^{2}$ distance as the scene matching score. If an imposter is detected by either the motion cue or the texture cue, then the system will refuse access. This method has some limitations for real-life applications, as the camera should be fixed and the background should not be monochrome.

The combination of motion and texture cues was widely used in competitions on countermeasures to 2D facial spoofing attacks [150] held respectively in 2011, 2013 and 2017. In the first competition [151] held in 2011, three of the six teams used multiple cues-based methods. The AMILAB team used jointly face motion detection, face texture analysis and eye-blinking detection in their solution (and the sum of weighted classification scores obtained by SVMs). The CASIA team also considered three different cues: motion cue, noise-based texture cue and face-background dependency cue. The UNICAMP team combined eye blinking, background-dependency and micro-texture of an image sequence. In the second competition [152] held in 2013, the two teams that obtained the best performances (CASIA and LNMIT) used multiple-cues based methods. CASIA proposed an approach based on the early (feature-based) fusion of motion and texture cues, whereas LNMIT combined LBP, 2D FFT and face–background consistency features [153] into a single feature vector, used as the input of Hidden Markov support vector machines (HM-SVMs) [154]. In the third competition [155] held in 2017, three teams used multiple cues-based methods. GRADIANT fused color [35], texture and motion information, exploiting both HSV and YCbCr color spaces. GRADIANT obtained the best performance on all protocols of the competition. Idiap proposed a score fusion method to fuse three cues (motion [147], texture (LBP) [37] and quality cue [32,64]) based on a Gaussian mixture model (GMM) to conduct face PAD. HKBU also proposed a multiple cue-based method fusing image quality [32], multi-scale LBP texture [55] and deep texture feature to give a robust presentation.

In 2016, Feng et al. [84] first integrated image-quality measures (see page 17) as a static texture cue and motion cues in a neural network. Three different cues, Shearlet-based image quality features (SBIQF) [156,157], a facial motion cue based on dense optical flow [158] and a scenic motion cue, were manually extracted and fed into the neural network (see Figure 24). The neural network had been pretrained and was then fine-tuned on the existing for face PAD datasets.

#### 2.5.2. Fusion of Liveness and 3D Geometric Cues

In 2018, Liu et al. [33] proposed to use CNN-Recurrent Neural Network (RNN) architecture for fusing the remote PhotoPlethysmoGraphy (rPPG) cue and the pseudo-depth map cue for face PAD (see Figure 25). This approach uses the fully connected CNN proposed in [36] to estimate the depth map (see page 23). Besides, a bypass connection is used to fuse the features from different layers, as in ResNet [4]. This work was the first one to proposed using RNN with LSTM units to learn the rPPG signal features based on the feature maps learned using CNNs. The estimation of the depth maps was calculated in advance using CNNs, whereas the depth maps’ ground truth was estimated in advance using [142,145,146] and the rPPG ground-truth was generated as described in Section 2.2.2. The authors also designed a non-registration layer to align the input face to a frontal face, as the input of the RNN, for estimating the rPPG signal features. Instead of designing a binary classifier, the face PAs are then detected by thresholding a score computed based on the weighted quadratic sum of the estimated depth map of the last frame of the video and the estimated rPPG signal features.

#### 2.5.3. Fusion of Texture and 3D Geometric Cues

In 2017, Atoum et al. [36] proposed to integrate patch-based texture cues and pseudo depth-map cues in two-stream CNNs for facial PAD as shown in Figure 26. The pseudo-depth map estimation, that aims at extracting the holistic features of an image, has been described in Section 2.4.2. The patch-based CNN stream (with 7 layers) focused on the image’s local features. The local patches, with fixed size, are randomly extracted from the input image. The label of a patch extracted from a real face is set to 1, whereas the label of the patch of extracted from a PA is set 0. Then, the randomly extracted patches with their labels are used to train the patch-based CNN stream with the softmax loss. Using the patch-level input not only increases the number of training samples but also forces the CNNs to explore the spoof-specific local discriminative information spreading in the entire face region. Finally, the two streams’ scores are weighed to sum up the final score to determine if the input image is a real face or a PA. As in [32,35,60], the authors also proposed to jointly use the HSV/YCbCr image with the RGB image as the input of the networks.

### 2.6. New Trends in PAD Methods

In this section, we describe the methods that constitute the leading edge of facial PAD methods based on RGB cameras. Thanks to the development of deep learning, especially in the computer vision domain, not only has face anti-spoofing detection performance been significantly boosted, but also many new ideas have been introduced. These new ideas relied on the following:the proposal of new cues to detect the face artifact (e.g., the pseudo-depth maps described in Section 2.4.2);learning the most appropriate neural networks architectures for facial PAD (e.g., using Neural Architecture Search (NAS) (see hereafter Section 2.6.1)); andaddress of the generalization issues, especially towards types of attacks that are not (or insufficiently) represented in the learning dataset. Generalization issues can be (at least partially) addressed using zero/few shot learning (see Section 2.6.2) and/or domain adaptation and adversarial learning (see Section 2.6.3).

The remainder of this section aims to present the two latter new trends more in details.

#### 2.6.1. Neural Architecture Search (NAS)-Based PAD Methods

In the last few years, deep neural networks have gained great success in many areas, such as speech recognition [159], image recognition [2,160] and machine translation [161,162]. The high performance of deep neural networks is heavily dependent on the adequation between their architecture and the problem at hand. For instance, the success of models like Inception [163], ResNets [4] and DenseNets [131] demonstrate the benefits of intricate design patterns. However, even with expert knowledge, determining which design elements to weave together generally requires extensive experimental studies [164]. Since the neural networks are still hard to design *a priori*, Neural Architecture Search (NAS) has been proposed to design the neural networks automatically based on reinforcement learning [165,166], evolution algorithm [167,168] or gradient-based methods [169,170]. Recently, NAS has been applied to several challenging computer vision tasks, such as face recognition [171], action recognition [172], person reidentification [173], object detection [174] and segmentation [175]. However, NAS has just started being applied to facial PAD.

In 2020, Yu et al. [82] first proposed to use NAS to design a neural network for estimating the depth map of a given RGB image for facial PAD. The gradient-based DARTS [169] and Pc-DARTS [170] search methods were adopted to search the architecture of cells forming the network backbone for facial PAD. Three levels of cells (low-level, mid-level and high-level) from the three blocks of CNNs in [33] (see Figure 25) were used for the search space. Each block has four layers, including three convolutional layers and one max-pooling layer, and is represented as a Directed Acyclic Graph (DAG), with each layer as a node.

The DAG is used to present all the possible connections and operations between the layers in a block, as shown in Figure 27. Instead of directly using the original convolutional layers as in [33], the authors proposed to use Central Difference Convolutional (CDC) layers, in which the sample values in local receptive field regions are subtracted to the value of the central position, similar to LBP. Then, the convolution operation is based on the local receptive field region with gradient values.

A Multiscale Attention Fusion Module (MAFM) is also proposed for fusing low-, mid- and high-level CDC features via spatial attention [82]. Finally, the searched optimal architecture of the networks for estimating the depth map of a given image is shown in Figure 28. Rather than [33] fusing multiple cues in the CNNs, this work only estimated the depth map of an input image to employ facial PAD by thresholding the mean value of the predicted depth map.

#### 2.6.2. Zero/Few-Shot Learning Based PAD Methods

Thanks to the significant development of deep learning, most state-of-the-art facial PAD methods show promising performance in intra-database tests on the existing public datasets [33,36,82] (see Section 4). Nevertheless, the generalization to cross-dataset scenarios is still challenging, in particular due to the possible presence in the test set of facial PAs that were not represented (or underrepresented) in the training dataset [32,65,155]. One possible solution is to collect a large-scale dataset to include as much diverse spoofing attack types as possible. However, as detailed below in Section 3, unlike other problems such as facial recognition where it is relatively easy to collect massive public dataset from the Internet, the images/videos of spoofing artifacts recaptured by a biometric system are quite rarely available on the Internet. Therefore, several research teams are currently investigating another solution that consists in leveraging zero/few-shot learning to detect the previously unseen face PAs. This problem has been named Zero-Shot Face Anti-spoofing (ZSFA) in [87].

In 2017, Arashloo et al. [86] first addressed unseen attack detection, as an anomaly detection problem where real faces constitute the positive class and are used to train a **one-class classifier** such as one-class SVM [176].

In the same spirit, in 2018, Nikisins et al. [85] also used one-class classification. However, they used one-class Gaussian Mixed Models (GMM) to model the distribution of the real faces in order to detect unseen attacks. Also, contrary to [86], they trained their model not only using one dataset but also aggregating three publicly available datasets (i.e., Replay-Attack [38], Replay-Mobile [38] and MSU MFSD [32], c.f. Section 3).

The abovementioned methods used only samples of genuine faces to train one-class classifiers, whereas in practice, known spoof attacks might also provide valuable information to detect previously unknown attacks.

This is why, in 2019, Liu et al. [87] proposed a CNN-based Deep Tree Network (DTN) in which 13 attack types covering both impersonation and obfuscation attacks were analyzed. First, they clustered the known PAs into eight semantic subgroups using unsupervised tree learning, and they used them as the eight leaf nodes of the DTN (see Figure 29). Then, Tree Routing Unit (TRU) was learned to route the known PAs to the appropriate tree leaf (i.e., subgroup) based on the features of known PAs learned by the tree nodes (i.e., Convolutional Residual Unit (CRU)). In each leaf node, a Supervised Feature Learning (SFL) module, consisting of a binary classifier and a mask estimator, was employed to discriminate between spoofing attacks. The mask estimation is similar to the depth map estimation as in the same authors’ previous work [33] (see page 12). Unseen attacks can then be discriminated based on the estimated mask and the score of a binary softmax classifier.

#### 2.6.3. Domain Adaption-Based PAD Methods

As detailed above, improving the generalization ability of existing facial PAD methods is one of the greatest challenges nowadays. To mitigate this problem, Pereira et al. [73] first proposed to combine multiple databases to train the model (see page 20). This is the most intuitive attempt towards improving generalization of the earned models. However, even by combining all the existing datasets, it is impossible to collect attacks from all possible domains (i.e., with every possible device and in all possible capture environments) to train the model. However, even though printed photo or video replay attacks from unseen domains may differ greatly from the source domain, they all are based on paper or video screens as PAI [88]. Thus, if there exists a generalized feature space underlying the observed multiple source domains and the (hidden but related) target domain, then domain adaptation can be applied [177,178].

In 2019, Shao et al. [88] first applied a domain adaption method based on adversarial learning [179,180] to tackle facial PAD. Under an adversarial learning schema, *N* discriminators were trained to help the feature generator produce generalized features for each of the *N* specific domains, as shown in Figure 30. Triplet loss is also used to enhance the learned generalized features to be even more discriminative, both within a database (intra-domain) and among different databases (inter-domain). To apply facial PAD, the learned generalized features are also trained to estimate the depth map of a given image as in [33] and to classify the image based on a binary classifier trained by softmax loss. This approach shows its superiority when increasing the number of source domains for learning generalized features. Indeed, in contrast, the previous methods without domain adaption such as LBP-TOP [71] or [33] cannot effectively improve the model’s generalization capacity, even when using multiple source datasets for training.

## 3. Existing Face Anti-Spoofing Datasets and Their Major Limitations

### 3.1. Some Useful Definitions

Facial PAD (anti-spoofing) datasets consist of two different kinds of documents (files) in the form of photos or videos:the set of “genuine faces”, that contains photos or videos of the genuine users’ faces (authentic faces of the alive genuine users), andthe set of “PA documents”, containing photos or videos of the PAI (printed photo, video replay, 3D mask, etc.)

Figure 31 illustrates the data collection procedure for constructing facial anti-spoofing databases.

In facial anti-spoofing datasets, genuine faces and PAIs (presented by imposters) are generally captured using the same device. This device plays the role of the biometric system’s camera in real-life authentication applications; we therefore chose to call it the “biometric system acquisition device”. For genuine faces and 3D mask attacks, only the biometric system acquisition device is used to capture the data.

However, for printed photo and video replay attacks, another device is used to create the PAI (photo or video) from a genuine face’s data. We call this device the “PA acquisition device”. It has to be noted that the PA acquisition device is in general different from the biometric system acquisition device. Some authors use the term “recapture”, as the original data is first collected using the PA acquisition device, then presented on a PAI and then recaptured using the biometric system acquisition device.

It has to be noted that, for photo display attacks and video replay attacks, the PAI itself can also be yet another electronic device. However, in general, only its screen is used, for displaying the PAI to the biometric system acquisition device. Of course, there could be datasets where the PAI also plays the role of the PA acquisition device, but this is not the case in general. Indeed, it is not the case in most real-life applications, where the imposter generally does not have control over the PA acquisition device (e.g., photos or videos found on the web).

For printed photo attacks, paper-crafted mask attacks and 3D mask attacks, yet another device is used: a printer. For photo attacks as well as paper-crafted mask attacks, usually (2D) printers are used. The printer’s characteristics as well as the quality of the paper used can greatly affect the quality of the PAI and therefore the chances of success of the attack. For 3D mask attacks, a 3D printer is used; its characteristics as well as the material used (e.g., silicone or hard resin) and its thickness also have an impact on the attack’s chances of success.

The devices used for each dataset’s collection are detailed in Table 2 and Section 3.4, together with a detailed description of these datasets. However, before that, in the remainder of this section, we successively give a brief overview of the existing datasets (Section 3.2) and describe their main limitations (Section 3.3).

### 3.2. Brief Overview of the Existing Datasets

The early studies of facial PAD, such as [42,44,45,46,56,83], are mostly based on private datasets. Such private datasets being quite limited, both in volume and diversity of attack types, makes it very difficult to fairly compare the different approaches.

The first public dataset was proposed in 2008 by Tan et al. [52]. The dataset was named NUAA, and it contains examples of photo attacks. The NUAA dataset enabled researchers to compare the results of their methods on the same benchmark. Later on, respectively in 2011 and 2012, Anjos et al. publicly shared the datasets PRINT-ATTACK [147] (containing photo attacks) and its extended version REPLAY-ATTACK [37] (containing video replay attacks as well). Quickly, these two datasets were widely adopted by the research community and as was one of the most challenging datasets: CASIA-FASD [19], that has also been published in 2012 and contains photo/video replay attacks but with more diversity in the PAs, PAIs and video resolutions. Later on, several other similar datasets were shared publicly for photo and video replay attacks, such as MSU-MFSD [32], MSU-USSA [66], OULU-NPU [181], SiW [33] and the very recent multi-modal dataset CASIA-SURF [182]. These datasets contain more diverse spoofing scenarios, such as MSU-MFSD [32], which first introduced a mobile phone scenario; MSU-USSA [66], which used celebrities’ photos from the Internet to increase the mass of data; OULU-NPU [181], which focused on attacks using mobile phones; and SiW [33], which contains faces with various poses, illumination and facial expressions.

The first public 3D mask attack dataset was 3DMAD [58], which includes texture maps, depth maps and point clouds together with the original images. Note that the depth maps and point clouds were collected by the Kinect 3D sensor rather than generic RGB cameras.

All the abovementioned datasets have been created under controlled environments, i.e., mostly indoors and with controlled illumination conditions, face poses, etc.

Although UAD [76] has collected videos from both indoors and outdoors and with a relatively large number of subjects, the dataset is no longer publicly available. Although MSU-USSA [66] contains a set of genuine faces captured under more diverse environments (including celebrities’ images collected from the Internet by [183]), the PAs always took place in controlled indoor conditions. Even the latest CASIA-SURF dataset [182], which is so far the largest multi-modal facial anti-spoofing dataset with 1000 subjects, contains only images collected in the same well-controlled conditions.

Therefore, public datasets are still far from reproducing real-world applications in a realistic way. This is probably due to the difficulty of collecting impostors’ PAs and PAIs in the wild. As a consequence, examples of PAs are generally acquired manually, which is a time-consuming and draining work. Creating a large-scale dataset for facial anti-spoofing in the wild, covering realistically various real-world applicative scenarios, is still a challenge. To circumvent these challenges, some researchers use data augmentation techniques to create synthetic (yet realistic) images of PAs [79].

A summarized overview of the existing public facial anti-spoofing datasets using only generic RGB cameras is provided in Table 2. More precisely, for each dataset, Table 2 gives (in columns) its release’s year (*Year*); the number of subjects it contains (*♯ Subj*); the ethnicity of the subjects in the dataset (*Ethnicity*); the type of PA represented in the dataset (*PA type(s)*); the number and type(s) of documents provided in the dataset as the cumulated number of genuine attempts and PAs (*Document ♯ & type(s)*); the PAI(s) used (*PAI*); the head pose(s) in the set of genuine faces (Pose); whether there are facial expression variations in the genuine faces dataset (*Expressions*); the biometric system acquisition device(s) for capturing both the genuine attempts and, in case of an attack, the PAI (*Biometric system acquisition device*); and the PA acquisition device that is possibly used to create the PAI (*PA acquisition device*).

### 3.3. Major Limitations of the Existing Datasets

Given the acquisition difficulties mentioned above, the existing face PAD datasets are (compared to other face-related problems) still limited not only in terms of volume but also in terms of diversity regarding the types of PAs, PAIs and acquisition devices used for genuine faces, PAs and possibly PAIs. In particular, as of today, there is still no public large-scale facial PAD in the wild, whereas there are several such datasets for facial recognition.

This hinders the development of effective facial PAD methods. It partly explains why, compared to other face-related problems, such as facial recognition, the performances of the current facial PAD methods are still below the requirements of most real-world applications (especially in terms of their generalization ability).

Of course, this is not the only reason: as detailed earlier in this paper, facial PAD is a very challenging problem. However, because all data-driven (learning-based) methods’ performances—including hand-crafted feature-based methods and more recent deep learning-based methods—are largely affected by the learning dataset’s volume and diversity [129,130,184,185], the lack of diversity in the datasets contribute to the limited performances of the current facial PAD methods.

More details about these datasets, including discussions about their advantages and drawbacks, are provided in the remainder of this section.

### 3.4. Detailed Description of the Existing Datasets

In this section, we provide a detailed description of all the datasets mentioned in Table 2.

NUAA Database [52] is the first publicly available facial PAD dataset for printed photo attacks. It includes some variability in the PAs, as the photos are moved/distorted in front of the PA acquisition device as follows:4 kinds of translations: vertical, horizontal, toward the sensor and toward the background2 kinds of rotations: along the horizontal axis and along the vertical axis (in-depth rotation)2 kinds of bending: along the horizontal and vertical axis (inward and outward)

A generic webcam is used for recording the genuine face images. Fifteen subjects were enrolled in the database, and each subject was asked to avoid eye blinking and to keep a frontal pose, with neutral facial expression. The attacks are performed by using printed photographs (either on photographic paper or A4 paper printed by a usual color HP printer). The dataset is divided into two separate subsets: for training and testing. The training set contains 1743 genuine face images and 1748 PAs impersonating 9 genuine users. The test set contains 3362 genuine samples and 5761 PAs. Viola-Jones detector [186] was used to detect the faces in the images, and the detected faces were aligned/normalized according to the eyes locations detected by [187]. The facial images were then resized to 64 × 64 pixels. Extracts from the NUAA database are shown in Figure 32.

PRINT-ATTACK Database [147] is the second proposed public dataset, including photo-attacks impersonating 50 different genuine users. The data was collected in two different conditions: controlled and adverse. In controlled conditions, the scene background is uniform and the light of a fluorescent lamp illuminates the scene, while in adverse conditions, the scene background is nonuniform and daylight illuminates the scene. A MacBook is used to record video clips of the genuine faces and the PAs. To capture the photos used for the attack, a 12.1 megapixel *Canon PowerShot SX150 IS* camera was used. These photos were then printed on plain A4 paper using a *Triumph-Adler DCC 2520* color laser printer. Video clips of about 10 s were captured for each PA under two different scenarios: hand-based attacks and fixed-support attacks. In hand-based attacks, the impostor held the printed photos using their own hands, whereas in fixed-support attacks, the impostors stuck the printed photos to the wall so they do not move/shake during the PA. Finally, 200 genuine attempts and 200 PA video clips were recorded. The 400 video clips were then divided into three subsets: training, validation and testing. Genuine identities (real identities or impersonated identities) in each subset were chosen randomly but with no overlap. Extracts of the PRINT-ATTACK dataset are shown in Figure 33.

CASIA-FASD Database [19] is the first publicly available face PAD dataset that provides both printed photo and video replay attacks. The CASIA-FASD database is a spoofing attack database which consists of three types of attacks: warped printed photos (which simulates paper mask attacks), printed photos with cut eyes and video attacks (motion cue such as eye blinking is also included). Each real face video and spoofing attack video is collected in three different qualities: low, normal and high quality. The high-quality video has a high resolution 1280 × 720, and the low/normal quality video has the same resolution 640 × 480. However, the low and normal quality is defined empirically by the perceptional feeling rather than strict quantitative measures. The whole database is split into a training set (containing 20 subjects) and a testing set (containing 30 subjects). Seven test scenarios are designed considering three different image qualities, three different attacks (warped/cut photo attack and video replay attack) and the overall test combining all the data. Examples of CASIA-FASD database are shown in Figure 34.

REPLAY-ATTACK Database [37] is an addendum of the abovementioned PRINT-ATTACK database [147] proposed by the same team. Compared to the PRINT-ATTACK database, REPLAY-ATTACK adds two more attacks, which are Phone-Attack and Tablet-Attack. The Phone-Attack uses an iPhone screen to display the video or photo attack, and the Tablet-Attack uses an iPad screen to display high-resolution (1024 × 768) digital photos or videos. Thus, the REPLAY-ATTACK database can be used to evaluate photo attacks using printed photo or screens, and video replay attacks. The number of video clips for spoof attacks is increased from 200 to 1000 for 50 identities (subjects). The dataset is divided into training, validation and test sets. REPLAY-ATTACK database also offers an extra subset as the enrollment videos for 50 genuine clients to be used for evaluating the vulnerabilities of a facial recognition system without facial PAD is vulnerable towards various types of attacks. Examples of REPLAY-ATTACK database are shown in Figure 35.

3DMAD Database [25,58] is the first public facial anti-spoofing database for 3D mask attacks. Previous databases contain attacks performed with 2D artifacts (i.e., photo or video) that are in general unable to fool facial PAD systems relying on 3D cues. In this database, the attackers wear customized 3D facial masks made out of a hard resin (manufactured by ThatsMyFace.com) of a valid user to impersonate the real access. It is worth mentioning that paper-craft mask files are also provided in this dataset. The dataset contains a total of 255 videos of 17 subjects. For each access attempt, a video was captured using the *Microsoft Kinect for Xbox 360*, which provides RGB data and depth information of size 640×480 at 30 frames per second. This dataset allows for the evaluation of both 2D and 3D PAD techniques, and their fusion. It is divided into three sessions: two real access sessions recorded with a time delay and one attack session captured by a single operator (attacker). Examples of the 3DMAD database are shown in Figure 36.

MSU-MFSD Database [32] is the first publicly available database to use mobile phones to capture real accesses. This database includes real access and attack videos for 55 subjects (among which 35 subjects are in the public version: 15 subjects in the training set and 20 subjects in test set). The genuine faces were captured using two devices: a Google Nexus 5 phone using its front camera (720×480 pixels) and a MacBook Air using its built-in camera (640×480 pixels). The Canon 550D SLR camera (1920 × 1088) and iPhone 5S (rear camera 1920 × 1080) are used to capture high-resolution pictures or videos (for photo attacks and video replay attacks). The printed high-resolution photo is played back using an iPhone 5S as PAI, and high definition (HD) (1920×1088) video-replays (captured on a Canon 550D SLR) are played back using an iPad Air. Examples of MSU-MFSD database are shown in Figure 37.

MSU-RAFS Database [59] is an extension of MSU-MFSD [32], CASIA-FASD [19] and REPLAY-ATTACK [37], where the video replay attacks are generated by replaying (on a MacBook) the genuine face videos in MSU-MFSD, CASIA-FASD and REPLAY-ATTACK. Fifty-five videos are genuine face videos from MSU-MFSD (captured by using the front camera of a Google Nexus 5), while 110 (2 × 55) videos are video replay attacks, captured using the built-in rear camera of a Google Nexus 5 and the built-in rear camera of an iPhone 6 and replayed using a MacBook as a PAI. In addition, 100 genuine face videos from CASIA-FASD and REPLAY-ATTACK were both used as genuine face videos and used to generate 200 video replay attacks by replaying these genuine face videos using a MacBook as a PAI. During the attack, the average standoff of the smartphone camera (used by the biometric system) from the screen of the MacBook was 15 cm, which assured that replay videos do not contain the bezels (edges) of the MacBook screen. Unlike the previously described databases, MSU-RAFS is constructed using existing genuine face videos (without having control over the biometric system acquisition devices used). Therefore, in this dataset, the biometric system acquisition devices used for capturing genuine face videos generally differ from the devices used for capturing the PAs. Thus, there is a risk of introducing bias when evaluating methods based on this dataset only.

Examples of MSU-RAFS database are shown in Figure 38.

UAD Database [76] is the first database to collect data both indoors and outdoors. It is also much bigger than the previous databases, both in terms of the number of subjects (440 subjects) and the number of videos (808 for training/16,268 for testing). All videos have been recorded at full-HD resolution, but subsequently cropped and resized to 1366×768 pixels. The dataset includes real access videos collected using six different cameras. For each subject, two videos are provided, both using the same camera but under different ambient conditions. Spoof attack videos corresponding to a given subject have also been captured using the same camera as for his/her real access videos. The video replay attacks have been displayed using seven different electronic monitors. However, this database seems to be no longer publicly available nowadays. Examples of UAD database are shown in Figure 39.

MSU-USSA Database [66] can be regarded as an extension of the MSU-RAFS [59], proposed by the same authors. There are two subsets in the database: (1) following the same idea as for MSU-RAFS, the first subset consists of 140 subjects from REPLAY-ATTACK [37] (50 subjects), CASIA-FASD [19] (50 subjects) and MSU-MFSD [32] (40 subjects); (2) the second subset consists of 1000 subjectstaken from the web faces database collected in [183], containing images of celebrities taken under a variety of backgrounds, illumination conditions and resolutions. Only a single frontal facial image of each celebrity is retained. Thus, the MSU-USSA database contains color facial images of 1140 subjects, where the average resolution of genuine face images is 705 × 865. Two cameras (front and rear cameras of a Google Nexus 5 smartphone) have been used to collect 2D attacks using four different PAIs (laptop, tablet, smartphone and printed photos), resulting in a total of 1140 genuine faces and 9120 PAs. Just like MSU-RAFS, MSU-USSA has not captured genuine face videos with the same device used for capturing the PAs. Thus, there is a risk of introducing bias when evaluating methods based on this dataset only. Examples of MSU-USSA database are shown in Figure 40.

OULU-NPU Database [181] is a more recent dataset (introduced in 2017) that contains PAD attacks acquired with mobile devices. In most previous datasets, the images were acquired in constrained conditions. On the other hand, this database contains a variety of motion, blur, illumination conditions, backgrounds and head poses. The database includes data corresponding to 55 subjects. The front cameras of 6 different mobile devices have been used to capture the images included in this dataset. The images have been collected under three separate conditions (environment/face artifacts/acquisition devices), each corresponding to a different combination of illumination and background. Presentation attacks include printed photo attacks created using two printers as well as video replay attacks using two different display devices. Four protocols are proposed for methods benchmarking (see Section 4.3 for more details). In total, the dataset is composed of 4950 real accesses and attack videos. Examples of the OULU-NPU database are shown in Figure 41.

SiW Database [33] is the first database to include facial spoofing attacks with both various labelled poses and facial expressions. This database consists of 1320 genuine access videos captured from 165 subjects and 3300 attack videos. Compared to the abovementioned databases, it includes subjects from a wider variety of ethnicities, i.e., Caucasian (35%), Indian (23%), African American (7%) and Asian (35%). Two kinds of print (photo) attacks and four kinds of video replay attacks have been included in this dataset. Video replay attacks have been created using four spoof mediums (PAIs): two smartphones, a tablet and a laptop. Four different sessions corresponding to different head poses/camera distances, facial expressions and illumination conditions were collected, and three protocols were proposed for benchmarking (see Section 4.3 for more details). Examples of the SiW database are shown in Figure 42.

CASIA-SURF Database [182] is currently the largest facial anti-spoofing dataset containing multi-modal images, i.e., RGB (1280 × 720), depth (640 × 480) and Infrared (IR) (640 × 480) images, of 1000 subjects in 21,000 videos. Each sample includes one live (genuine) video clip and six spoof (PA) video clips under different types of attacks. Six different photo attacks are included in this database: flat/warped printed photos where different regions are cut from the printed face. During the dataset capture, genuine users and imposters were required to turn left or right, to move up or down and to walk towards or away from the camera (imposters holding the printed color photo on an A4 paper). The face angle was only limited to 300 degrees. Imposters stood within a range of 0.3 to 1.0 m from the camera. The RealSense SR300 camera was used to capture the RGB, depth and Infrared (IR) images. The database is divided into three subsets for training, validation and testing. In total, there are 300 subjects and 6300 videos (2100 for each modality) in the training set, 100 subjects and 2100 videos (700 for each modality) in the validation set, and 600 subjects and 12,600 videos (4200 for each modality) in the testing set. Examples of the CASIA-SURF database are shown in Figure 43.

## 4. Evaluation

In this section, we present a comprehensive evaluation of the approaches for facial PAD detailed in Section 2. By doing so, our objective is to investigate the strengths and weaknesses of the different types of methods in order to draw future research directions for facial PAD to make facial authentication less vulnerable to imposters. We first present (in Section 4.1) the evaluation protocol, then (in Section 4.2) the evaluation metrics, and finally (in Section 4.3) the comparison of the results reported in the reviewed works.

### 4.1. Evaluation Protocol

In this section, we present the protocol we used to compare experimentally the different facial PAD methods. In the early studies of facial anti-spoofing detection, there was no uniform protocol to train and evaluate the facial PAD methods. In 2011, a first standard protocol was proposed by Anjos et al. [147] in order to fairly compare the different methods. In 2017, a second standard protocol was proposed by Boulkenaf et al. [181] based on an ISO/IEC standard. On top of the evaluation metrics to be used, these protocols address mainly two aspects: (1) how to divide the database and (2) what kinds of tests should be conducted for evaluation, e.g., intra-database and inter (cross)-database tests.

#### (a) Dataset division

The protocol proposed by Anjos et al. [147] is widely used when the evaluation is based on PRINT-ATTACK [147], REPLAY-ATTACK [37], OULU-NPU [181], 3DMAD [25,58] and/or CASIA-SURF [182]. This protocol relies on the division of the dataset into three subsets: training, validation set and test sets (respectively for training, tuning the model’s parameters and assessing the performances of the tuned model).

Other databases, such as CASIA-FASD [19], MSU-MFSD [32] and SiW [33], only consist of two independent subsets: training and test subsets. In this case, either a small part of the training set is used as a validation set or cross-validation is used to tune the model’s parameters. In some datasets (such as OULU-NPU and SiW), the existence of different sessions explicitly containing different capture conditions allowed the authors to propose refined protocols for evaluation (as detailed above in Section 3.4) and can also be used for dataset division.

#### (b) Intra-database vs. inter-database evaluation

An intra-database evaluation protocol uses only a single database to both train and evaluate the PAD methods. However, as the current databases are still limited in terms of variability, intra-database evaluations can be subject to overfitting, and therefore, report biased (optimistic) results.

The inter-database test, proposed by Pereira et al. [73] for evaluating the generalization abilities of a model, consists in training the model on a certain database and then evaluating it on a separate database. Although inter-database tests (or cross-database tests) aim to evaluate the model’s generalization abilities, it is important to note that the evaluation performances are still affected by the distribution of the two datasets. Indeed, if the two datasets’ distribution is close (e.g., if the same PAIs or spoof acquisition devices were used), the inter-database test will also report optimistic results. For instance, for a given model, inter-database evaluations between PRINT-ATTACK and MSU-MFSD (using the same MacBook camera to acquire the images) result in much better performances than inter-database evaluations between CASIA-FASD and MSU-MFSD (where CASIA-FASD uses a USB camera with very differentfrom a MacBook), as reported in [32].

### 4.2. Evaluation Metric

To compare different PAD methods, Anjos et al. [147] proposed in 2011 to use Half Total Error Rate (*HTER*) as an evaluation metric. As a PAD system is subject to two types of errors, either the real accesses are rejected (false rejection) or the attacks are accepted (false acceptance); *HTER* combines the False Rejection Rate (*FRR*) and the False Acceptance Rate (*FAR*) to measure the PAD performance as follows:(1)$$HTER=\frac{FRR+FAR}{2}$$
where the *FAR* and *FRR* are respectively defined as
(2)$$FAR=\frac{FP}{FP+TN}$$
(3)$$FRR=\frac{FN}{FN+TP}$$
with *TP*, *FP*, *TN* and *FN* respectively corresponding to the numbers of true positives (the accepted real accesses), false positives (the accepted attacks), true negatives (the rejected attacks) and false negatives (the rejected real accesses). *TP*, *FP*, *TN* and *FN* are calculated using model parameters based on a selected threshold achieving Equal Error Rate (EER) on the validation set (the selected threshold for which FRR = FAR). It can be noted that EER is also often used for assessing the model’s performance on the validation and training subsets.

However, since 2017 and the work proposed in [181], the performance is most often reported using the metrics defined in the standardized ISO/IEC 30107-3 metrics [16]: Attack Presentation Classification Error Rate (*APCER*) and Bona Fide Classification Error Rate (*BPCER*) (also called Normal Presentation Classification Error Rate (*NPCER*) in some research papers). These two metrics correspond respectively to the False Acceptance Rate (*FAR*) and the False Rejection Rate (*FRR*), but for obtaining *APCER*, the *FAR* is computed separately for each PAI/type of attack and *APCER* is defined as the highest *FAR* (i.e., the *FAR* of the most successful type of attack). Similar to *HTER*, the Average Classification Error Rate (*ACER*) is then defined as the mean of *APCER* and *BPCER* using the model parameters achieving EER on the validation set: (4)$$ACER=\frac{APCER+NPCER}{2}$$

On top of the *HTER* and *ACER* scalar values, the Receiver Operating Characteristic (ROC) curve and the Area Under the Curve (AUC) are also commonly used to evaluate the PAD method’s performance. The latter two have the advantage that they can provide a global evaluation of the model’s performances over different values of the parameter set.

### 4.3. Comparison and Evaluation of the Results

In this part, we compare some of the facial PAD methods detailed in Section 2 on the public benchmarks presented in Section 3 following the intra-database and inter/cross-database protocols described above. Among the more than 50 methods presented in Section 2, we selected here the most influential methods and/or the ones that are among the most characteristic of their type of approach (following the typology presented in Section 2.1 and Figure 3, page 7 and used for the methods presentation in Section 2). To compare the performances of the different methods, we used the metrics EER, HETR, APCER, BPCER and ACER described above. The results we report are extracted from the original papers introducing the methods, i.e., we did not redevelop all these methods to perform the evaluation ourselves. As a consequence, some values might be missing (and are then noted as “–”) in the following tables. Another point that is important to note is that we chose to focus our analysis on the type of features used. Of course, depending on the method, the type of classifier used (or the neural network architecture, for end-to-end deep learning methods) might also have an impact on the overall performance. However, we consider that, for each method, the authors have chosen to use the most effective classifier/architecture and that therefore the overall performances they report are largely representative of the descriptive and discriminative capabilities of the features they use.

#### 4.3.1. Intra-Database Evaluation on Public Benchmarks

Table 3 and Table 4 respectively show the results of the intra-database evaluation on the CASIA-FASD and REPLAY-ATTACK datasets. Compared to static texture feature-based methods such as DoG, LBP-based methods, dynamic texture-based methods such as LBP-TOP, Spectral Cubes [77] and DMD [23] are more effective on both benchmarks. However, the static features learned using CNNs can boost the performance significantly and sometimes even outperform dynamic texture hand-crafted features. For instance, even the earliest CNN-based method with static texture feature [34] has shown a superior performance in terms of HTER than almost all previously introduced state-of-the-art methods based on hand-crafted features. It was the first time that deep CNNs showed potential for facial PAD. Later, researchers proposed more and more models learning static or dynamic features based on deep CNNs such as LSTM-CNN [78], DPCNN [68] and Patch-based CNN [36], that has achieved the state-of-the-art performances on both the CASIA-FASD and REPLAY-ATTACK datasets. Besides, we can see both in Table 3 and Table 4 that Patch-Depth CNN [36], fusing different cues (i.e., texture cue and 3D geometric cue (depth map)), has shown its superiority over single cue-based methods using the same CNN, such as Patch cue-based CNN [36] or Depth cue-based CNN [36]. Indeed, their HTERs are respectively 2.27%, 2.85% and 2.52% on CASIA-FASD and 0.72%, 0.86% and 0.75% on REPLAY-ATTACK. These results show the effectiveness of multiple cues-based methods that, by leveraging different cues, are able to effectively detect a wider variety of PA types.

As explained earlier in Section 3.4, on OULU-NPU and SiW, the different protocols corresponding to different applicative scenarios were proposed.

More specifically, for the benchmark OULU-NPU, four protocols were proposed in [181]:Protocol 1 aims to test the PAD methods under different environmental conditions (illumination and background);Protocol 2’s objective is to test the generalization abilities of the methods learnt using different PAIs;Protocol 3 aims to test the generalization across the different acquisition devices (i.e., using Leave One Camera Out (LOCO) protocol to test the method over six smartphones); andProtocol 4 is the most challenging scenario, as it combines the three previous protocols to simulate real-world operational conditions.

For SiW, three different protocols were proposed in [33]:Protocol 1 deals with variations in facial pose and expression;Protocol 2 tests the model over different spoof mediums (PAIs) for video replay; andProtocol 3 tests the methods over different PAs, e.g., learning from photo attacks and testing on video attacks and vice versa.

From Table 5 and Table 6, one can see that, both for OULU-NPU and SiW and for all the evaluation protocols, the best methods are the 3D geometric cue methods using depth estimation. Furthermore, the architectures obtained using NAS with depth maps (e.g., CDCN++ [82]) has achieved state-of-the-art performances both on OULU-NPU and SiW.

Moreover, we can see that the protocols used on OULU-NPU for testing generalization abilities (protocols 2 and 3) are especially challenging. When considering the protocol defined to evaluate the performances in near real-worldapplicative conditions (protocol 4), the model’s performance can degrade up to 25 times compared to “easier” protocols (e.g., CDCN++’s ACER arises from 0.2% for protocol 1 to 5.0% for protocol 4). Similar results can be observed on SiW.

It indicates that the generalization across scenarios is still a challenge for facial PAD methods, even within the same dataset.

#### 4.3.2. Cross-Database Evaluation on Public Benchmarks

Compared to the promising results shown in the intra-database test, the inter/cross-database test results are still way worse than most real-world applications requirements. Several databases have been adopted to perform cross (inter)-database evaluation, such as CASIA-FASD vs. MSU-MFSD [32] and MSU-USSA vs. REPLAY-ATTACK/CASIA-FASD/MSU-MFSD [66]. However, most researchers have reported their cross-database evaluation results using REPLAY-ATTACK vs. CASIA-FASD [33,69,73,79,81,82], since the important differences between these two databases introduce a great challenge for cross-database testing.

Table 7 reports the results of cross-database tests between REPLAY-ATTACK and CASIA-FASD. Although the use of deep learning methods significantly improves the generalization between different databasets, there is still a large gap compared to the intra-database results. Especially if we train the model on REPLAY-ATTACK and then test the trained model on CASIA-FASD, even the best methods can only achieve at best a 29.8% HTER.

Moreover, all the PAD methods based on hand-crafted features show weak generalization abilities. For instance, the HTER of LBP-based methods based on RGB image (such as basic LBP [37] and LBP-TOP [73]) are about 60%. However, the LBP in HSV/YCbCr color space shows a comparable or even better generalization ability than some deep learning-based methods (e.g., the method in [60] achieves 30.3% and 37.7% HTER when trained on CASI-FASD and REPLAY-ATTACK, respectively). It is noteworthy that the multiple cues-based method Auxiliary [33], by fusing depth map and rPPG cues, achieves a good generalization even in the most difficult cross-database tests. For instance, when trained on REPLAY-ATTACK and tested on CASIA-FASD, it achieves slightly better HTER (28.4%) than the latest method CDCN++ [82] based on NAS (29.8%, see Table 7). This demonstrates that the multiple cues-based methods, when using different cues that are inherently complementary to each other, can achieve better generalization than facial PAD models based on single cues. However, from a very general perspective, improving the generalization abilities of the current face PAD methods is still a great challenge for facial anti-spoofing.

## 5. Discussion

From the evaluation results presented in the previous Section 4.3, we can see that facial PAD is still a very challenging problem. In particular, the performances of the current facial PAD methods are still below the requirements of most real-world applications (especially in terms of generalization ability).

More precisely, the performances are acceptable when there is not too much variation between the conditions of the genuine faces capture for enrollment and the genuine face/PA presentation for authentication (intra-database evaluation).

However,

all hand-crafted features show a limited generalization ability, as they are not powerful enough to capture all the possible variations in the acquisition conditions; andthe features learned by deep/wide neural networks are of very high dimensions, compared to the limited size of the training data.

Thus, both types of features suffer from overfitting and therefore poor generalization capacity.

Therefore, learning features that are able to discriminate between a genuine face and any kind of PA, possibly under very different capture conditions, is still an open issue. This issue will be discussed in Section 5.1. Then, in Section 5.2, we discuss a less studied topic in the field of facial PAD: how to detect obfuscation attacks.

### 5.1. Current Trends and Perspectives

As stated earlier, learning features that are distinctive enough to discriminate between genuine faces and various PAs, possibly in very different environments, is still an open issue. Of course, this kind of issue (related to the generalization abilities of data-driven models) are common in the field of computer vision, way beyond facial PAD.

However, as collecting impostors’ PAs and PAIs is nearly impossible in the wild, collecting/creating a face PAD dataset with sufficient samples and variability (not only regarding the capture conditions but also regarding the different types of PA/PAIs) is still very time-consuming and costly (see Section 3.2). It is indeed much easier to create a dataset for most object recognition tasks (e.g., face authentication).

In order to tackle all previously seen PAs, a current trend is to combine multiple cues (see Section 2.5). However, due to the abovementioned challenges in the dataset creation as well as the technological advances that ill-intentioned users can access to deploy increasingly sophisticated attacks, the PAD method might have to detect PAs that were not included in its training dataset. This problem, called “Unknown attack” previously, is especially challenging.

Beyond the current methods that try and use zero/few-shot learning approaches to tackle this problem, the question of learning features that are representative enough of “real” faces, so that they can discriminate between genuine faces and any kind of PAs under any type of capture condition, is still an open issue. This issue, for which some researchers have recently proposed solutions based on domain adaptation, will very probably raise a lot of attention from researchers in the coming years, especially with the emergence of ever more sophisticated DeepFakes.

### 5.2. Obfuscation Face PAD

As stated in Section 1.2, two types of PAs are defined in the relevant ISO standard [16]: impersonation (spoofing) attacks, i.e., attempts of impostors to impersonate a genuine user, and obfuscation attacks, i.e., attempts for the impostor to hide her own identity.

Most current facial PAD research focuses on the former type, i.e., impersonation spoofing, as it is the most frequent attack for biometric systems based on facial recognition/authentication.

However, there are some applicative scenarios where obfuscation attacks are very important to detect. For instance, in law-enforcement applications based on video surveillance, one of the main objectives is to be able to detect criminals, whereas the goals of criminals using obfuscation attacks is to remain unrecognized by the system.

As detailed in Figure 1 on page 3, obfuscation attacks may entail (possibly extreme) facial makeup or occluding significant portions of the face using scarves, sunglasses, face masks, hats, etc. In some cases, the person deploying obfuscation attack may also use tricks that are usually used for impersonation attacks, e.g., by using a mask showing the face of a noncriminal. As well as impersonation attacks, obfuscation attacks also include previously unseen attacks/unknown attacks. The detection of unseen/unknown obfuscation attacks is still an open issue which needs to be further studied. 

To the best of our knowledge, so far, the only dataset containing examples of obfuscation attacks is SiW-M. This dataset has been introduced in [87], where the authors have shown the effectiveness of extreme makeup for facial obfuscation. One solution is to process the facial image so as to synthetically "remove" the makeup, as in [190,191].

More generally, given that compared to impersonation attacks obfuscation attacks are still less frequent, several research groups consider obfuscation attacks as a zero/few-shot PAD problem [87].

Even though obfuscation attack detection has been so far much less studied than impersonation attack detection, it is very likely that this topic will become more and more studied in the future, given the conjunction of several factors, such as the generalization of video surveillance in public places, geo-political issues including risks of terrorist attacks in some regions of the world and recent technological developments that allow researchers to tackle this problem.

## 6. Conclusions

In this survey paper, we have thoroughly investigated over 50 of the most influential face PAD methods that can work in scenarios where the user only has access to RGB cameras of Generic Consumer Devices. By structuring our paper according to a typology of facial PAD methods based on the types of PA that they are aiming to thwart and in chronological order, we have shown the evolution in the facial PAD during the last two decades. This evolution covers a large variety of methods, from hand-crafted features to the most recent deep learning-based technologies such as Neural Architecture Search (NAS). Benefiting from the recent breakthroughs obtained by researchers in computer vision, thanks to the advent of deep learning, facial PAD methods are getting ever more effective and efficient. We have also gathered, summarized and detailed the most relevant information about a dozen of the most widespread public datasets for facial PAD.

Using these datasets as benchmarks, we have extensively compared different types of facial PAD methods using common experimental protocol and evaluation metrics. This comparative evaluation allows us to point out which types of approaches are most effective, depending on the type of PA. More specifically, according to our investigation, texture-based methods which are also the most widely used PAD methods, and especially dynamic texture-based methods, are able to detect almost all types of PAs. Furthermore, the methods based on texture features learned using deep learning have significantly improved the state-of-the-art facial PAD performances compared to methods based on hand-crafted texture features. However, in general, high-quality 3D mask attacks are still a great challenge for texture-based approaches. On the other hand, liveness-based methods or 3D geometric-based methods can achieve relatively better generalization capabilities, even though they are still vulnerable to video replay attacks or complex illumination conditions. Multiple cues-based methods, by leveraging different cues for facial PAD, are in general more effective for detecting various PAs. Nevertheless, the computational complexity of multiple-cues based methods is an issue that needs to be considered for real-time applications. Partly because of the complexity of the facial PAD problem, the huge variability in the possible attacks and the lack of dataset that contains enough samples with sufficient variability, all current approaches are still limited in terms of generalization.

We have also identified some of the most prominent current trends in facial PAD, such as combining approaches that aim to thwart various kinds of attacks or to tackle previously unseen attacks. We have also provided some insights for future research and have listed the still open issues, such as learning features that are able to discriminate between genuine faces and all kinds of PAs.

## Figures and Tables

**Figure 1 jimaging-06-00139-f001:**
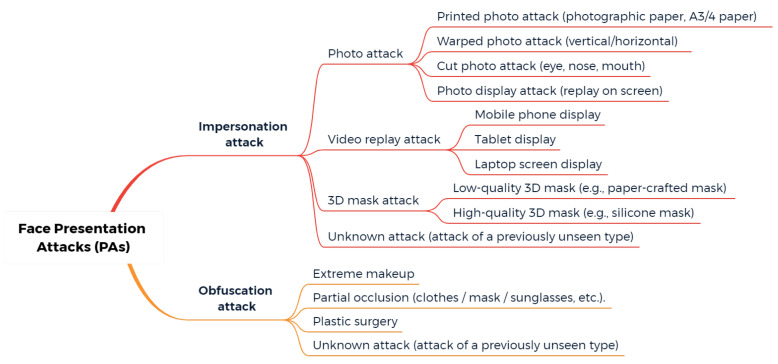
A typology of facial Presentation Attacks (PAs).

**Figure 2 jimaging-06-00139-f002:**
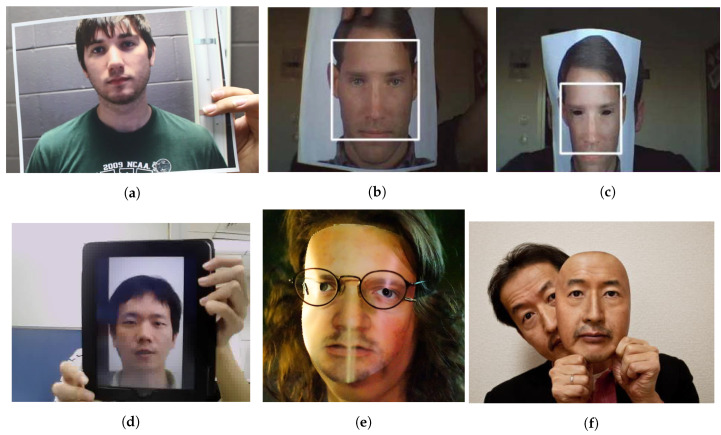
Examples of common facial presentation attacks: (**a**) a printed photo attack from the SiW dataset [17]; (**b**) an example of a warped photo attack extracted from [18]; (**c**) an example of a cut photo attack extracted from [18]; (**d**) a video replay attack from the CASIA-FASD dataset [19]; (**e**) a paper-crafted mask from UMRE [20]; and (**f**) a high-quality 3D mask attack from REAL-f [17].

**Figure 3 jimaging-06-00139-f003:**
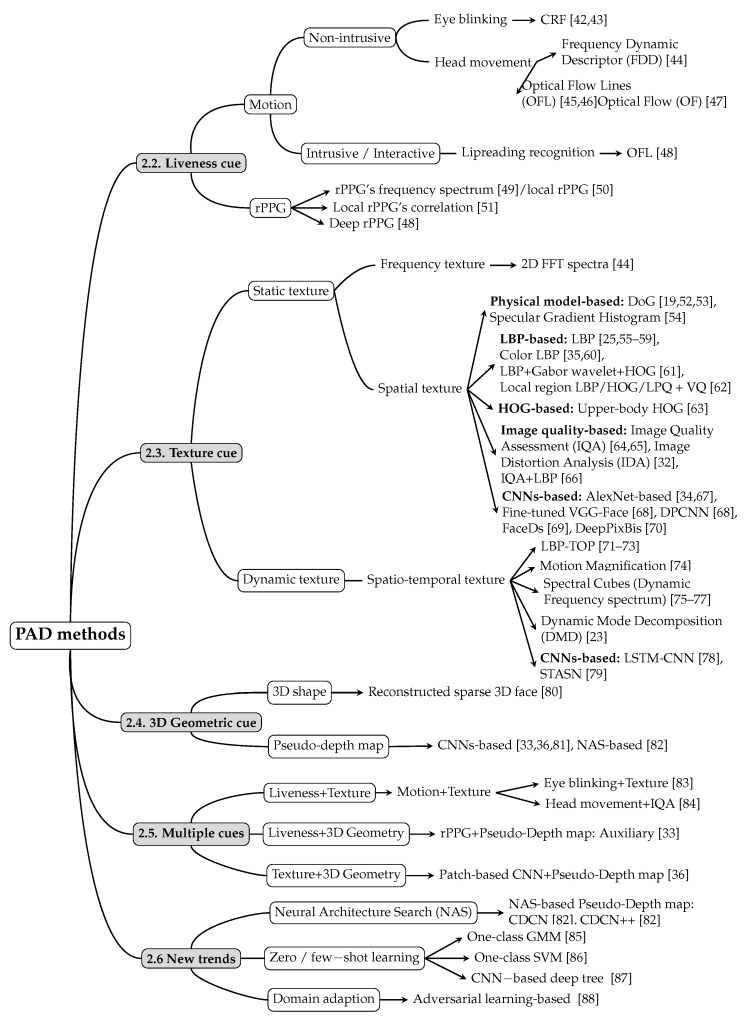
Our proposed typology for facial PAD methods [19,23,25,32,33,34,35,36,42,43,44,45,46,47,48,49,50,51,52,53,54,55,56,57,58,59,60,61,62,63,64,65,66,67,68,69,70,71,72,73,74,75,76,77,78,79,80,81,82,83,84,85,86,87,88]: the darkest nodes are numbered with their corresponding subsections in the remainder of Section 2.

**Figure 4 jimaging-06-00139-f004:**
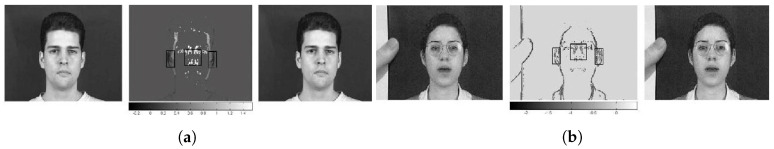
The Optical Flow Lines (OFL) images obtained from (**a**) a genuine (alive face) presentation with a subtle facial rotation and (**b**) a printed photo attack with horizontal translation [46]: in (**a**), the inner parts are brighter than the outer parts of the face, which is characteristic of the motion differences between different face parts. In contrast, all parts of the planar photo in (**b**) display constant motion.

**Figure 5 jimaging-06-00139-f005:**
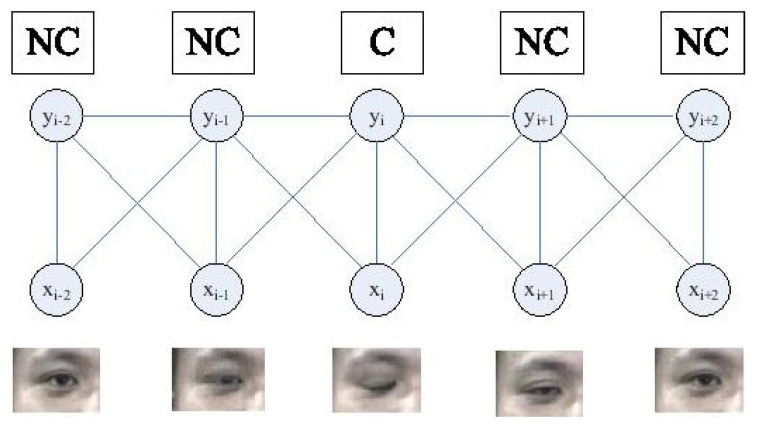
Conditional Random Field (CRF)-based eye-blinking model [42,43]: in this example, each hidden state $y_{i}$ is conditioned by its corresponding observation (image) $x_{i}$ and its two neighboring observations $x_{i-1}$ and $x_{i+1}$.

**Figure 6 jimaging-06-00139-f006:**
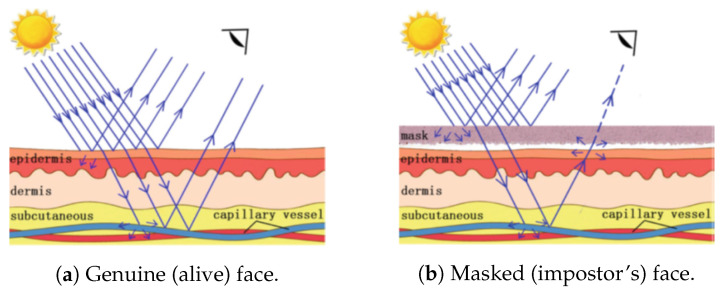
Illustration of how Remote PhotoPlethysmoGraphy (rPPG) can be used to detect blood flow for facial PAD [17]: (**a**) on a genuine (alive) face, the light penetrates the skin and illuminates capillary vessels in the subcutaneous layer. Blood oxygen saturation changes within each cardiac cycle, leading to periodic variations in the skin’s absorption and reflection of the light. These variations are observable by RGB cameras. (**b**) On a masked face, the mask’s material blocks light absorption and reflection, leading to no (or insignificant) variations in the reflected light.

**Figure 7 jimaging-06-00139-f007:**
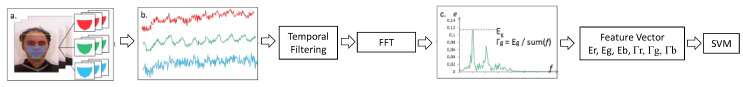
Framework of the rPPG-based method proposed in [49] (Figure extracted from [49]): (**a**) the ROI for extracting the rPPG signal; (**b**) the extracted rPPG signal for each RGB channel; (**c**) the frequence spectrum for calculating frequency features.

**Figure 8 jimaging-06-00139-f008:**
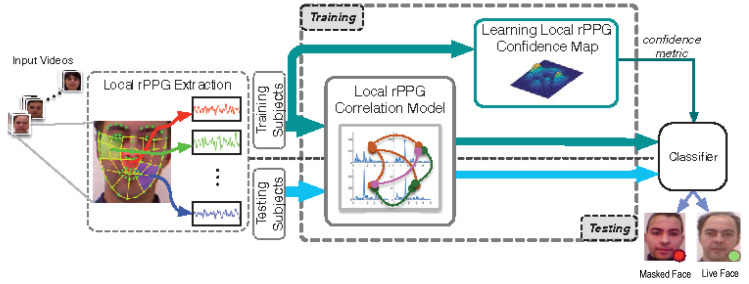
Framework of the local rPPG correlation-based method proposed in [51] (Figure extracted from [51]).

**Figure 9 jimaging-06-00139-f009:**
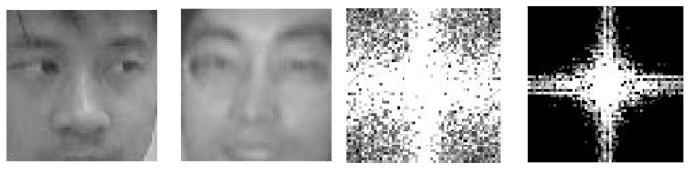
From left to right: a genuine (alive) face, a printed photo attack and their respective 2D Fourier spectra (Figure extracted from [44]).

**Figure 10 jimaging-06-00139-f010:**
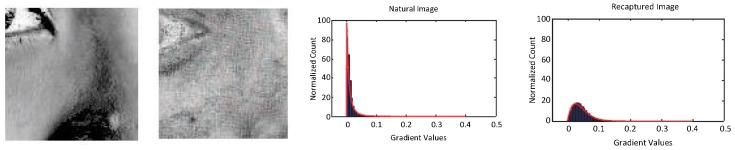
From **left** to **right**: patches extracted from a genuine face image and a printed photo, and their respective specular gradient histograms (Figure extracted from [54]).

**Figure 11 jimaging-06-00139-f011:**
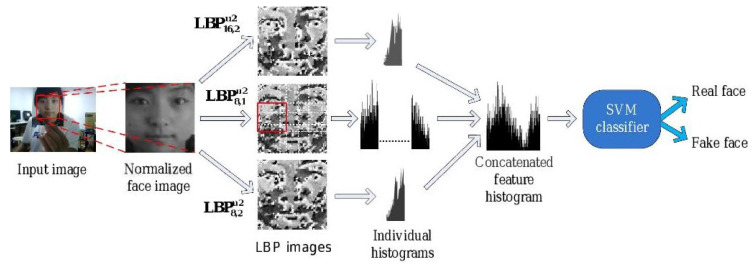
Illustration of the approach proposed in [55]: firstly, the face is detected, cropped and normalized into a 64×64 pixel image. Then, $LBP_{8,2}^{u2}$ and $LBP_{16,2}^{u2}$ are applied on the normalized face image, which generates a 59-bin histogram and a 243-bin histogram respectively. The obtained $LBP_{8,1}^{u2}$ image is also divided into 3×3 overlapping regions (as shown in the middle row). As each region generates a 59-bin histogram, a single 531-bin histogram is obtained by their concatenation. Then, all individual histograms are concatenated to obtain a 833-bin/dimension (59 + 243 + 531) histogram, which is fed to a nonlinear SVM classifier to detect photo/video replay attacks.

**Figure 12 jimaging-06-00139-f012:**
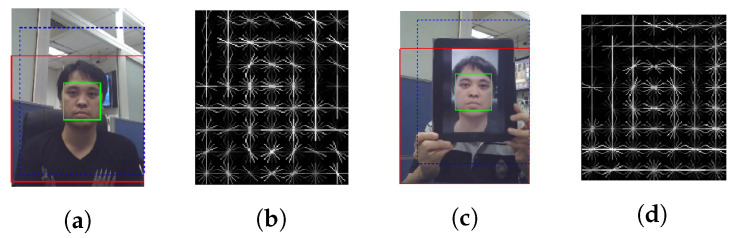
Examples of upper-body images from the CASIA-FASD dataset [19] and their Histogram of Oriented Gradient (HOG) features [63]: (**a**) upper-body of a genuine face; (**b**) HOG feature of the blue dashed rectangle in (**a**); (**c**) video replay attack; and (**d**) HOG feature of the blue dashed rectangle in (**c**). The figure was extracted from [63].

**Figure 13 jimaging-06-00139-f013:**
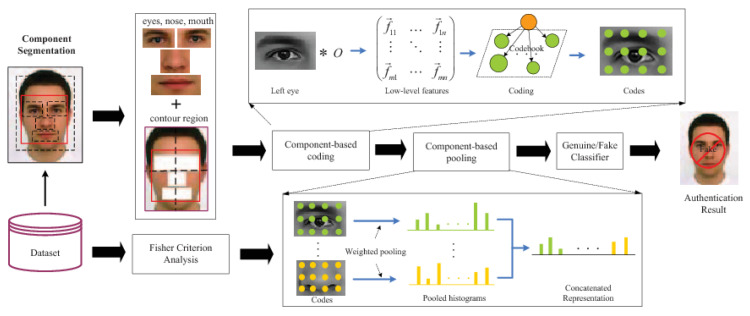
Illustration of the approach proposed in [62] (Figure extracted from [62]).

**Figure 14 jimaging-06-00139-f014:**
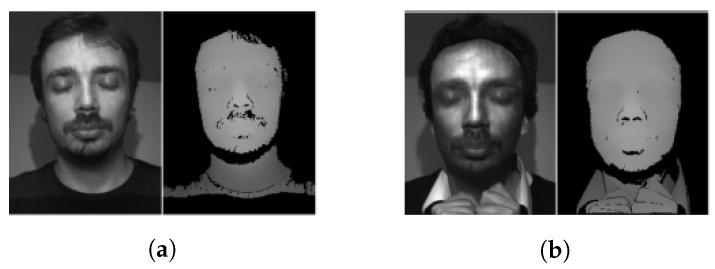
The texture image and its corresponding depth image for (**a**) a real access and (**b**) a 3D mask attack: the figure was extracted from [25].

**Figure 15 jimaging-06-00139-f015:**
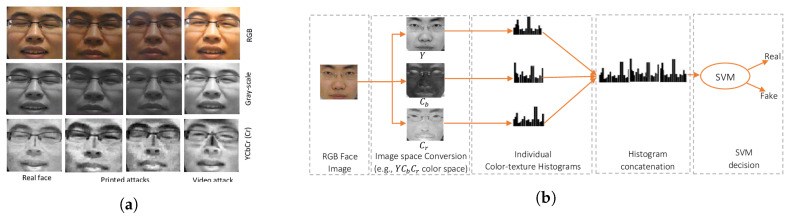
Illustration of the method in [60]: (**a**) an example of a genuine face presentation and its corresponding printed photo and video attacks in RGB, greyscale and YCbCr colour space and (**b**) the architecture of the method proposed in [60].

**Figure 16 jimaging-06-00139-f016:**
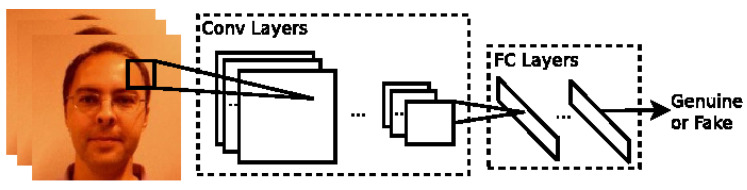
Illustration of the method proposed in [34] (Figure extracted from [34]).

**Figure 17 jimaging-06-00139-f017:**
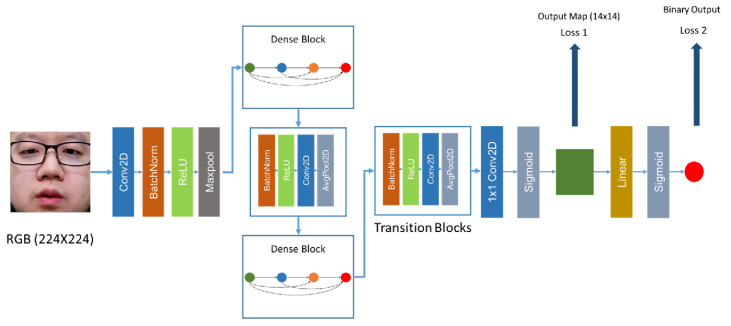
The diagram of Deep Pixelwise Binary Supervision (DeepPixBiS) as shown in [70].

**Figure 18 jimaging-06-00139-f018:**
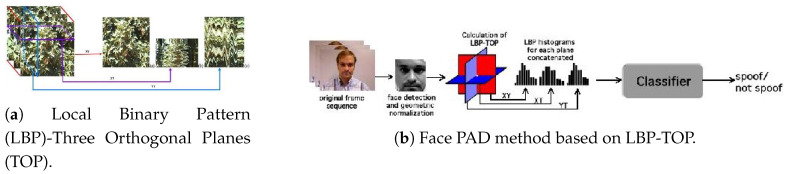
Illustration of the LBP-TOP from [117] and LBP-TOP-based facial PAD method [71]: (**a**) the three orthogonal planes, i.e., XY plane, XT plane and YT plane, of LBP-TOP features extracted from an image sequence and (**b**) the framework of the approach based on LBP-TOP introduced in [71].

**Figure 19 jimaging-06-00139-f019:**
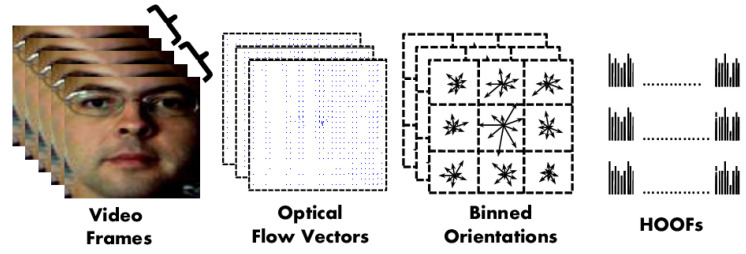
Illustration of the Histogram of Oriented Optical Flows (HOOF) feature proposed in [74].

**Figure 20 jimaging-06-00139-f020:**
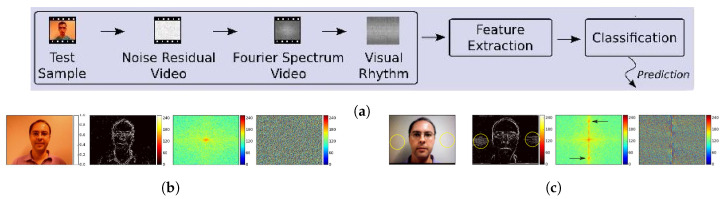
Illustration of the noise residual video and the visual rhythm-based approach as shown in [75,77]: (**a**) the framework of the visual rhythm-based approach, (**b**) example of valid access and (**c**) example of a frame from a video replay attack. For (**b**,**c**), from left to right: original frame, residual noise frame, magnitude spectrum and phase spectrum. In (**c**), the yellow circles in the original image and its corresponding residual noise frame highlight the Moiré effect. The black arrows in the magnitude spectrum show the impact of the Moiré effect on the Fourier spectrum.

**Figure 21 jimaging-06-00139-f021:**
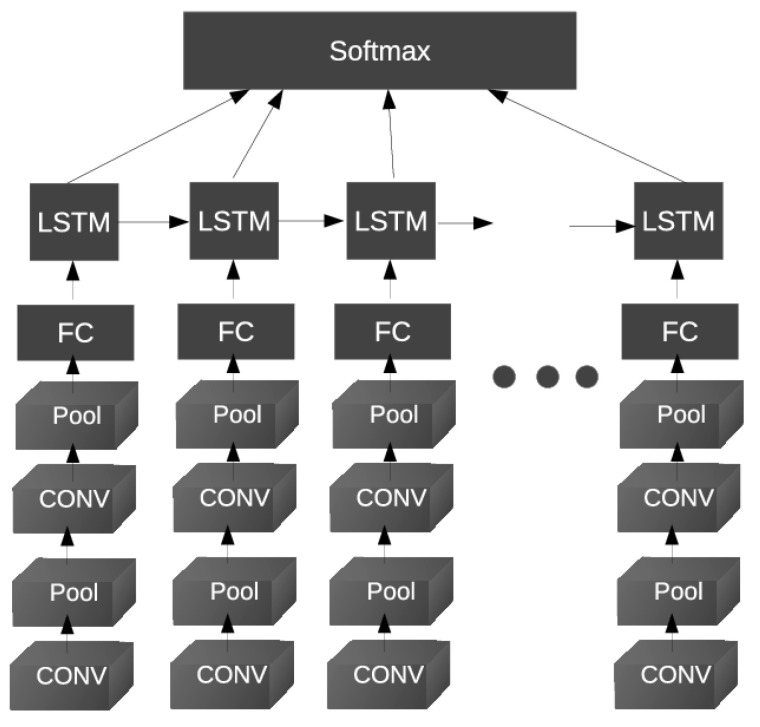
Long Short-Term Memory (LSTM)-Convolutional Neural Network (CNN) architecture used in [78] for facial PAD.

**Figure 22 jimaging-06-00139-f022:**
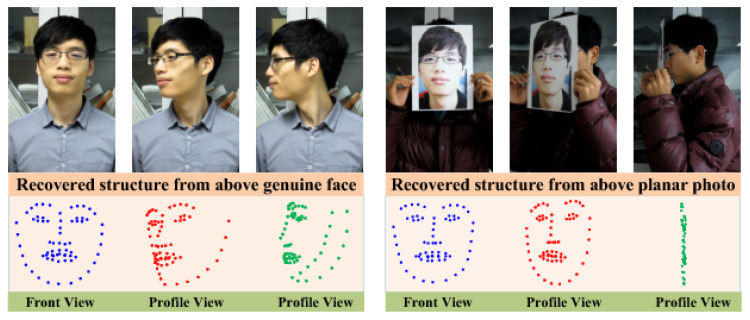
Illustration of reconstructed sparse 3D structures of genuine face (**left**) and photo attack (**right**) [80].

**Figure 23 jimaging-06-00139-f023:**
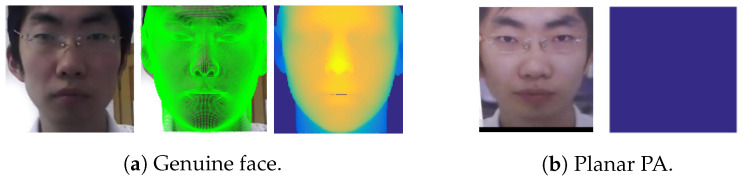
(**a**) A real face image, the fitted 3D face model and the depth map of the real 3D face, and (**b**) a planar PA and its ground-truth depth map: the yellow/blue colors in the depth map represent respectively a closer/further point to the camera. The figure was extracted from [36].

**Figure 24 jimaging-06-00139-f024:**
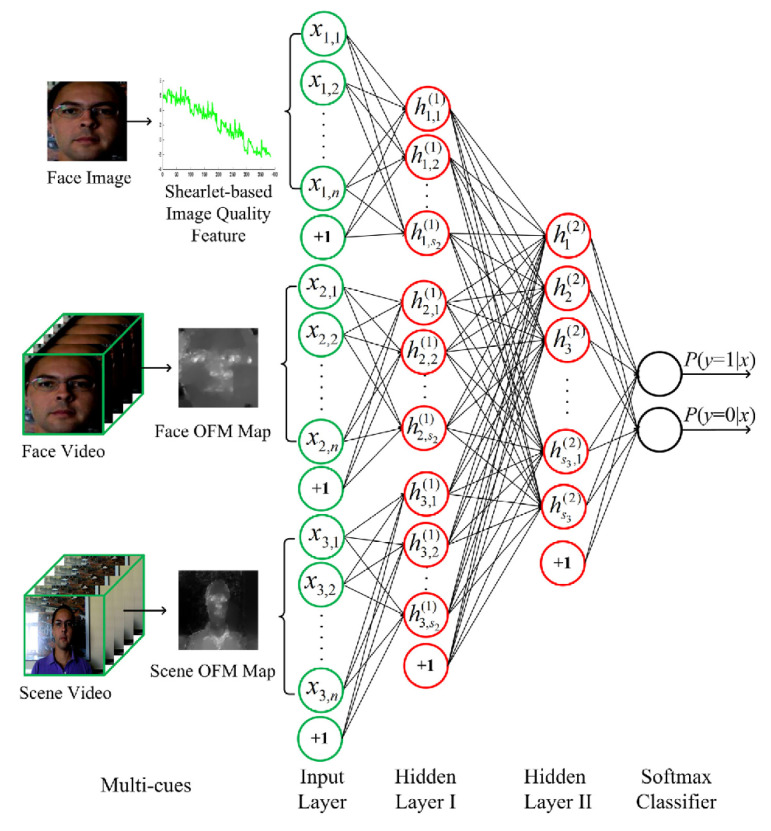
A flowchart of the multiple cues-based face PAD method using neural networks as shown in [84].

**Figure 25 jimaging-06-00139-f025:**
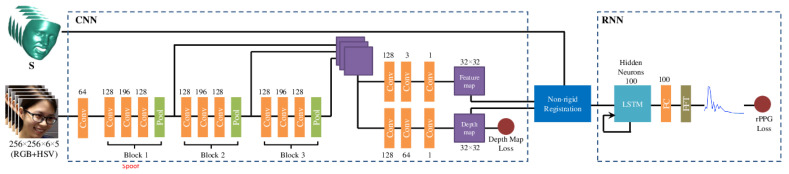
The proposed rPPG and depth-based CNN-RNN architecture for facial PAD as shown in [33].

**Figure 26 jimaging-06-00139-f026:**
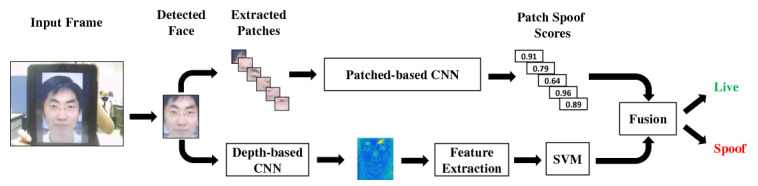
The architecture of the proposed patch-based and depth-based CNNs for facial PAD as shown in [36].

**Figure 27 jimaging-06-00139-f027:**
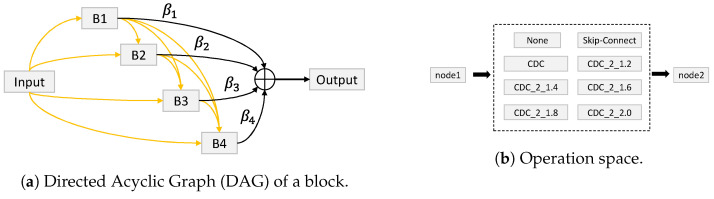
The search space of Neural Architecture Search (NAS) for forming the network backbone for face PAD as shown in [82]: (**a**) Directed Acyclic Graph (DAG) of a block. Each node represents a layer in the block, and the edge is the possible information flow between layers. (**b**) Operation space, listing the possible operations between layers (8 operations were defined in [82]).

**Figure 28 jimaging-06-00139-f028:**
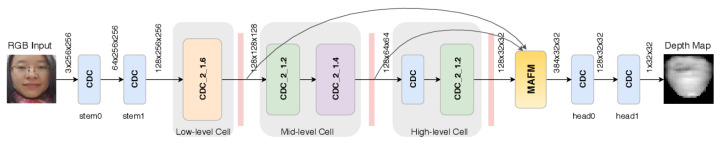
The architecture of the NAS-based backbone for depth map estimation as shown in [82].

**Figure 29 jimaging-06-00139-f029:**
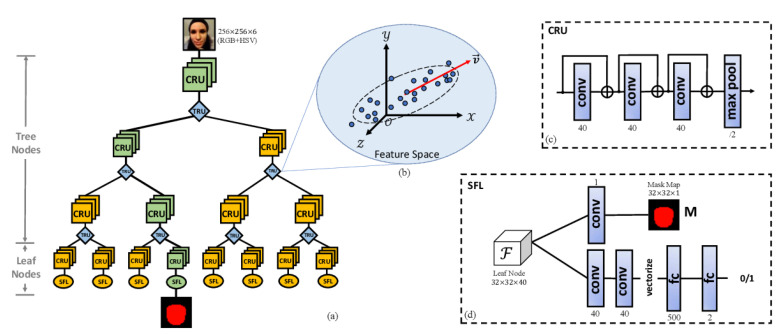
The proposed Deep Tree Network (DTN) architecture as shown in [87]: (**a**) overall structure of the DTN. A tree node consists of a Convolutional Residual Unit (CRU) and a Tree Routing Unit (TRU), whereas a leaf node consists of a CRU and a Supervised Feature Learning (SFL) module. (**b**) Tree Routing Unit (TRU) assigns the feature learned by CRU to a given child node based on an eigenvalue analysis similar to Principal component analysis (PCA). (**c**) Structure of each Convolutional Residual Unit (CRU). (**d**) Structure of the Supervised Feature Learning (SFL) in the leaf nodes.

**Figure 30 jimaging-06-00139-f030:**
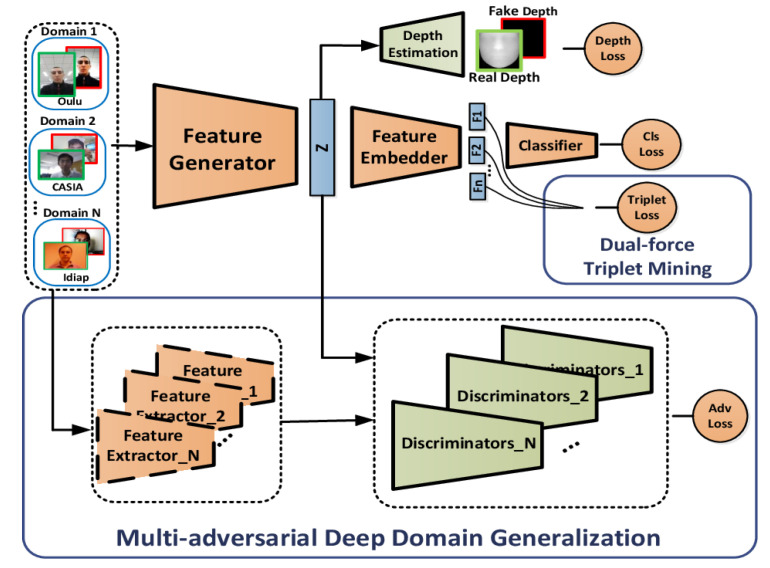
Overview of the domain adaption approach based on adversarial learning for facial anti-spoofing, as shown in [88]: each discriminator is trained to help the generalized features (learned by the generator from multiple source domains) to better generalize on their corresponding source domain. The depth loss, triplet loss and the classification loss (“Cls loss”) are then used to enhance the ability to discriminate any kind of PA.

**Figure 31 jimaging-06-00139-f031:**
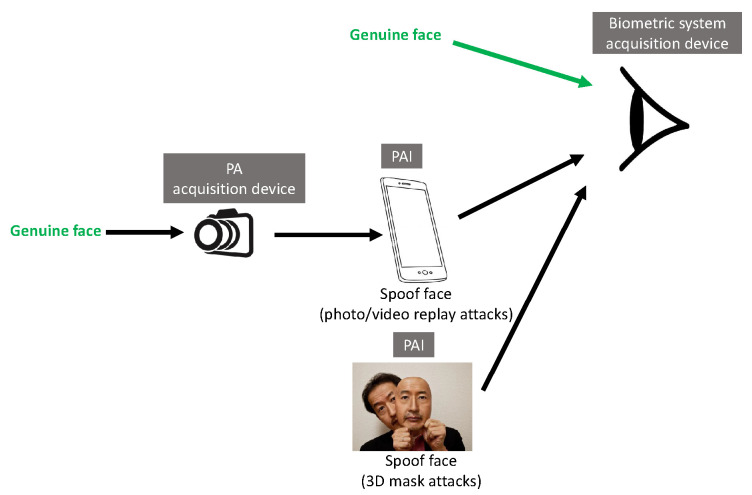
Diagram illustrating the data collection procedure for constructing facial anti-spoofing databases.

**Figure 32 jimaging-06-00139-f032:**
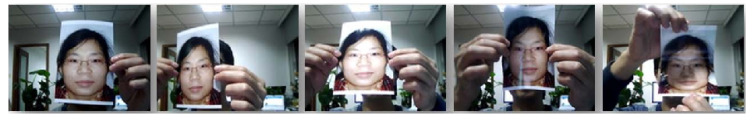
The NUAA Database (from left to right): five different photo attacks.

**Figure 33 jimaging-06-00139-f033:**
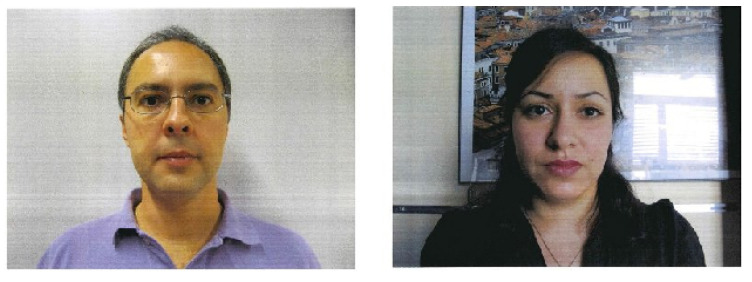
PRINT-ATTACK (from **left** to **right**): photo attack under controlled and adverse scenarios.

**Figure 34 jimaging-06-00139-f034:**
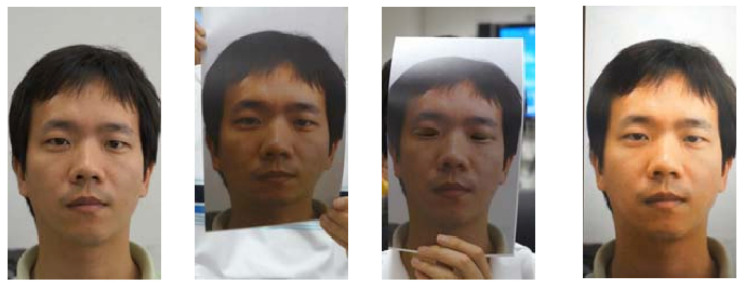
CASIA-FASD (from **left** to **right**): real face, two warped/cut photo attacks and a video replay attack.

**Figure 35 jimaging-06-00139-f035:**
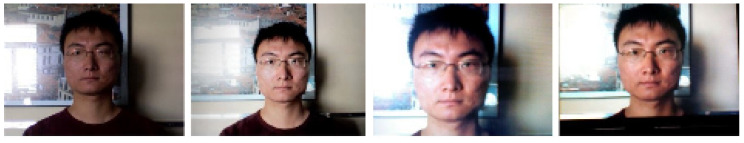
REPLAY-ATTACK (from **left** to **right**): real face, video replay attack, photo displayed on screen and printed photo attack.

**Figure 36 jimaging-06-00139-f036:**
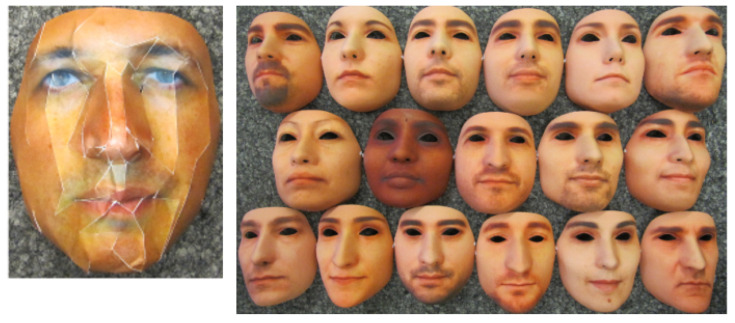
3DMAD (from **left** to **right**): paper-craft mask and 17 hard resin masks.

**Figure 37 jimaging-06-00139-f037:**
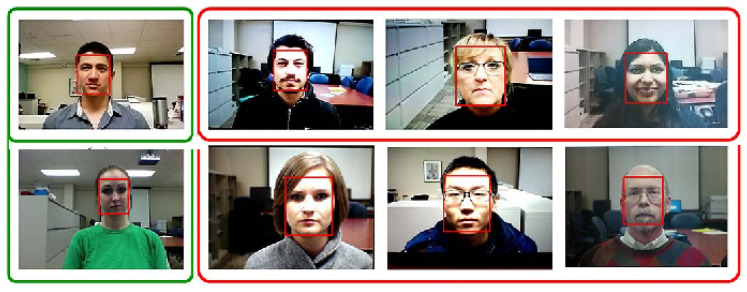
MSU-MFSD (from **left** to **right**): genuine face, video replay attacks respectively displayed on iPad and iPhone, and printed photo attack.

**Figure 38 jimaging-06-00139-f038:**
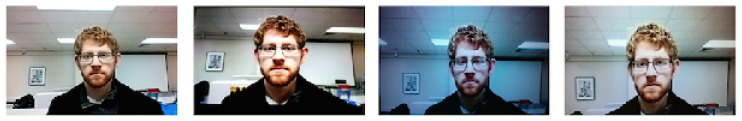
MSU-RAFS (from **left** to **right**): genuine face and PAs from MSU-MFSD (attacks using a MacBook (as a PAI) to replay the genuine attempts from MSU-MFSD captured by different devices (respectively Google Nexus 5 and iPhone 6)).

**Figure 39 jimaging-06-00139-f039:**
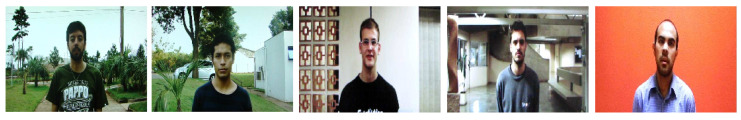
UAD (from **left** to **right**): video replay attacks, captured outdoors (first and second images) and indoors (last three images).

**Figure 40 jimaging-06-00139-f040:**
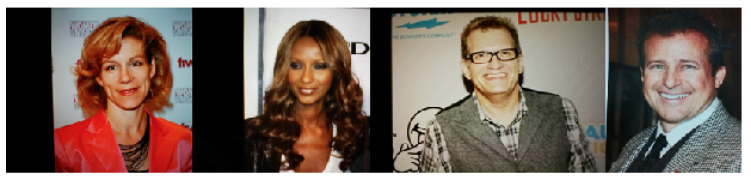
MSU-USSA (from **left** to **right**): spoof faces recaptured from the celebrity dataset [183].

**Figure 41 jimaging-06-00139-f041:**
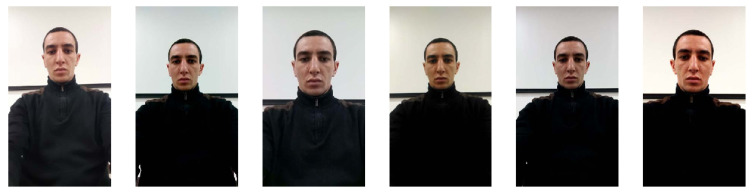
Extracts of the OULU-NPU dataset.

**Figure 42 jimaging-06-00139-f042:**
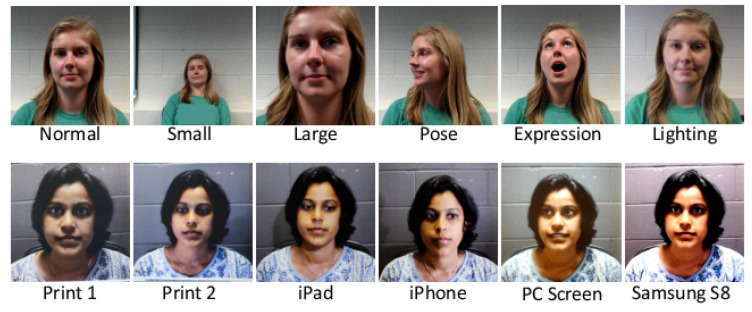
SiW: genuine access (**top**) and PA (**bottom**) videos with different poses, facial expressions and illumination conditions for genuine accesses and PAI devices for PAs.

**Figure 43 jimaging-06-00139-f043:**
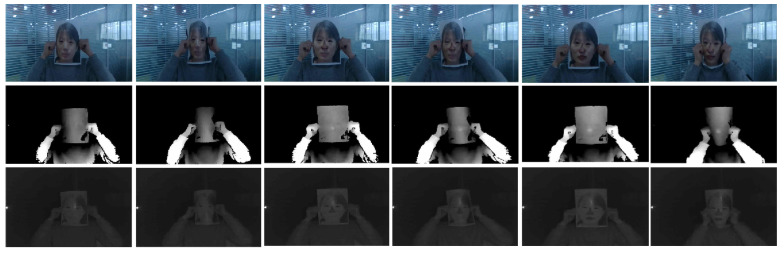
Extract from CASIA-SURF showing six photo attacks in each 3 modalities: RGB (**top**), depth (**middle**) and IR (**bottom**).

**Table 1 jimaging-06-00139-t001:** The type(s) of Presentation Attacks (PAs) each subtype that the PAD method aims to detect.

PAD Methods	Subtypes	PAs
Liveness cue-based	Nonintrusive motion-based	Photo attack (except cut photo attack)
	Intrusive motion-based	Photo attack (except cut photo attack)
		Video replay attacks (except sophisticated DeepFakes)
	rPPG-based	Photo attack
		“Low quality” video replay attacks
		3D mask attack (low/high quality)
Texture cue-based	Static texture-based Dynamic texture-based	Photo attack
		Video replay attack
		3D mask attack (low quality)
3D Geometry cue-based	3D shape-based Pseudo-depth map-based	Photo attack Video replay attack
Multiple cues-based	Liveness (Motion) + Texture	Photo attack
		Video replay attack
	Liveness + 3D Geometry (rPPG + Pseudo-depth map)	Photo attack
		Video replay attack
		3D mask attack (low/high quality)
	Texture + 3D Geometry (Patched-base texture + Pseudo-depth map)	Photo attack Video replay attack

**Table 2 jimaging-06-00139-t002:** A summary of public facial anti-spoofing datasets based on generic RGB camera.

Database	Year	♯ Subj.	Ethnicity	PA Type(s)	Document ♯ & Type(s) Images (I)/Videos (V)	PAI	Pose	Expression	Biometric System Acquisition Device	PA Acquisition Device
NUAA [52]	2010	15	Asian	Printed photos Warped photos	5105/7509 (I)	A4 paper	Frontal	No	Webcam (640 × 480)	Webcam (640 × 480)
PRINT-ATTACK [147]	2011	50	Caucasian	Printed photos	200/200 (V)	A4 paper	Frontal	No	Macbook Webcam (320 × 340)	Cannon PowerShot SX150 (12.1 MP)
CASIA-FASD [19]	2012	50	Asian	Printed photos Warped photos Cut photos Video replay	200/450 (V)	Copper paper iPad 1 (1024 × 768)	Frontal	No	Sony NEX-5 (1280 × 720) USB Camera (640 × 480)	Sony NEX-5 (1280 × 720) Webcam (640 × 480)
REPLAY-ATTACK [37]	2012	50	Caucasian 76% Asian 22% African 2%	Printed photos Photo display 2× video replays ^a^	200/1000 (V)	A4 paper iPad 1 (1024 × 768) iPhone 3GS (480 × 320)	Frontal	No	Macbook Webcam (320 × 340)	Canon PowerShot SX 150 (12.1MP) iPhone 3GS
3DMAD [25,58]	2013	17	Caucasian	2× 3D masks ^b^	170/85 (V)	Paper-crafted mask Hard resin mask (ThatsMyFace.com)	Frontal	No	Kinect (RGB camera) (Depth sensor)	—
MSU-MFSD [32]	2015	35	Caucasian 70% Asian 28% African 2%	Printed photos 2× video replays	110/330 (V)	A3 paper iPad Air (2048 × 1536) iPhone 5s (1136 × 640)	Frontal	No	Nexus 5 (built-in camera software 720 × 480) Macbook Air (640 × 480)	Cannon 550D (1920 × 1088) iPhone 5s (1920 × 1080)
MSU-RAFS [59]	2015	55	Caucasian 44% Asian 53% African 3%	Video replays	55/110 (V)	Macbook (1280 × 800)	Frontal	No	Nexus 5 (rear: 3264 × 2448) iPhone 6 (rear: 1920 × 1080)	The biometric system acquistion devices used in MSU-MFSD, CASIA-FASD, REPLAY-ATTACK.
UAD [76]	2015	404	Caucasian 44% Asian 53% African 3%	7× video replays	808/16,268 (V)	7 display devices	Frontal	No	6 different cameras (no moible phone) (1366 × 768)	6 different cameras (no moible phone) (1366 × 768)
MSU-USSA [66]	2016	1140	Diverse set (from web faces database from the [183])	Printed photos Photo display 3× video replays	1140/9120 (V)	White paper (11 × 8.5 paper) Macbook (2080 × 1800) Nexus 5 (1920 × 1080) Tablet (1920 × 1200)	Frontal	Yes	Nexus 5 front: 1280 × 960) (rear: 3264 × 2448) iPhone 6 (rear: 1920 × 1080)	Same as MSU-RAFS Cameras used to capture celebrities’ photos are unknown.
OULU-NPU [181]	2017	55	Caucasian 5% Asian 95%	Printed photos 2× video replays	1980/3960 (V)	A3 glossy paper Dell display (1280 × 1024) Macbook (2560 × 1600)	Frontal	No	Samsung Galaxy S6 (rear: 16 MP)	Samsung Galaxy S6 (front: 5 MP) HTC Desire EYE (front: 13 MP) MEIZU X5 (front: 5 MP) ASUS Zenfone Selfi (front: 13 MP) Sony XPERIA C5 (front: 13 MP) OPPO N3 (front: 16 MP)
SiW [33]	2018	165	Caucasian 35% Asian 35% African American 7% Indian 23%	Printed photos (high/low-quality photos) 4× video replays	1320/3300 (V)	Printed paper (High/low-quality) Samsung Galaxy S8 iPhone 7 iPad Pro PC screen(Asus MB168B)	[$-90^{\circ },90^{\circ }$]	Yes	Camera (1920 × 1080)	Camera (1920 × 1080) Camera (5184 × 3456)
CASIA-SURF [182]	2019	1000	Asian	Flat-cut/Warped-cut photos (eyes, nose, mouth)	3000/18,000 (V)	A4 paper	[$-30^{\circ },30^{\circ }$]	No	RealSense (RGB camera) (1280 × 720) (Depth sensor) (640 × 480) (IR sensor) (640 × 480)	RealSense (RGB camera) (1280 × 720) (Depth sensor) (640 × 480) (IR sensor) (640 × 480)

^a^*x* × video replays denotes *x* types of video replay attacks with different PAIs; ^b^ 2 × 3D masks denotes two types of 3D masks attacks: paper-crafted masks and hard resin masks.

**Table 3 jimaging-06-00139-t003:** Evaluation of various facial PAD methods on CASIA-FASD.

Method	Year	Feature	Cues	EER (%)	HTER (%)
DoG [19]	2012	DoG	Texture (static)	17.00	-
LBP [37]	2012	LBP	Texture (static)	-	18.21
LBP-TOP [72]	2014	LBP	Texture (dynamic)	10.00	-
Yang et al. [34]	2014	CNN	Texture (static)	4.92	4.95
Spectrual Cubes [77]	2015	FourrierSpectrum +codebook	Texture (dynamic)	14.00	-
DMD [23]	2015	LBP	Texture (dynamic)	21.80	-
Color texture [35]	2015	LBP	Texture (HSV/static)	6.20	-
LSTM-CNN [78]	2015	CNN	Texture (dynamic)	5.17	5.93
Color LBP [60]	2016	LBP	Texture (HSV/static)	3.20	-
Fine-tuned VGG-Face [68]	2016	CNN	Texture (static)	5.20	-
DPCNN [68]	2016	CNN	Texture (static)	4.50	-
Patch-based CNN [36]	2017	CNN	Texture (static)	4.44	3.78
Depth-based CNN [36]	2017	CNN	Depth	2.85	2.52
Patch-Depth CNN [36]	2017	CNN	Texture+Depth	2.67	2.27

**Table 4 jimaging-06-00139-t004:** Evaluation of various facial PAD methods on REPLAY-ATTACK.

Method	Year	Feature	Cues	EER (%)	HTER (%)
LBP [37]	2012	LBP	Texture (static)	13.90	13.87
Motion Mag [74]	2013	HOOF	Texture (dynamic)	-	1.25
LBP-TOP [72]	2014	LBP	Texture (dynamic)	7.88	7.60
Yang et al. [34]	2014	CNN	Texture (static)	2.54	2.14
Spectral Cubes [77]	2015	Fourier Spectrum +codebook	Texture (dynamic)	-	2.80
DMD [23]	2015	LBP	Texture (dynamic)	5.30	3.80
Color texture [35]	2015	LBP	Texture (HSV/static)	0.40	2.90
Moire pattern [59]	2015	LBP+SIFT	Texture (static)	-	3.30
Color LBP [60]	2016	LBP	Texture (HSV/static)	0.10	2.20
Fine-tuned VGG-Face [68]	2016	CNN	Texture (static)	8.40	4.30
DPCNN [68]	2016	CNN	Texture (static)	2.90	6.10
Patch-based CNN [36]	2017	CNN	Texture (static)	2.50	1.25
Depth-based CNN [36]	2017	CNN	Depth	0.86	0.75
Patch-Depth CNN [36]	2017	CNN	Texture+Depth	0.79	0.72

**Table 5 jimaging-06-00139-t005:** Evaluation of various facial PAD methods on OULU-NPU.

Protocol	Method	Year	Feature	Cues	APCER (%)	BPCER (%)	ACER (%)
**1**	CPqD [155]	2017	Inception-v3 [188]	Texture (static)	2.9	10.8	6.9
**1**	GRADIANT [155]	2017	LBP	Texture (HSV/dynamic)	1.3	12.5	6.9
**1**	Auxiliary [33]	2018	CNN+LSTM	Depth+rPPG	1.6	1.6	1.6
**1**	FaceDs [69]	2018	CNN	Texture (Quality/static)	1.2	1.7	1.5
**1**	STASN [79]	2019	CNN+Attention	Texture (dynamic)	1.2	2.5	1.9
**1**	FAS_TD [81]	2019	CNN+LSTM	Depth	2.5	0.0	1.3
**1**	DeepPixBis [70]	2019	DenseNet [131]	Texture	0.8	0.0	0.4
**1**	CDCN [82]	2020	CNN	Depth	0.4	1.7	1.0
**1**	CDCN++ [82]	2020	NAS+Attention	Depth	0.4	0.0	0.2
**2**	MixedFASNet [155]	2017	DNN	Texture (HSV/static)	9.7	2.5	6.1
**2**	GRADIANT [155]	2017	LBP	Texture (HSV/dynamic)	3.1	1.9	2.5
**2**	Auxiliary [33]	2018	CNN+LSTM	Depth+rPPG	2.7	2.7	2.7
**2**	FaceDs [69]	2018	CNN	Texture (Quality/static)	4.2	4.4	4.3
**2**	STASN [79]	2019	CNN+Attention	Texture (dynamic)	4.2	0.3	2.2
**2**	FAS_TD [81]	2019	CNN+LSTM	Depth	1.7	2.0	1.9
**2**	DeepPixBis [70]	2019	DenseNet [131]	Texture (static)	11.4	0.6	6.0
**2**	CDCN [82]	2020	CNN	Depth	1.5	1.4	1.5
**2**	CDCN++ [82]	2020	NAS+Attention	Depth	1.8	0.8	1.3
**3**	MixedFASNet [155]	2017	DNN	Texture (HSV/static)	5.3±6.7	5.30±6.7	5.3±6.7
**3**	GRADIANT [155]	2017	LBP	Texture (HSV/dynamic)	2.6±3.9	5.0±5.3	3.8±2.4
**3**	Auxiliary [33]	2018	CNN+LSTM	Depth+rPPG	2.7±1.3	3.1±1.7	2.9±1.5
**3**	FaceDs [69]	2018	CNN	Texture (Quality/static)	4.0±1.8	3.8±1.2	3.6±1.6
**3**	STASN [79]	2019	CNN+Attention	Texture (dynamic)	4.7±3.9	0.9±1.2	2.8±1.6
**3**	FAS_TD [81]	2019	CNN+LSTM	Depth	5.9±1.9	5.9±3.0	5.9±1.0
**3**	DeepPixBis [70]	2019	DenseNet [131]	Texture	11.7±19.6	10.6±14.1	11.1±9.4
**3**	CDCN [82]	2020	CNN	Depth	2.4±1.3	2.2±2.0	2.3±1.4
**3**	CDCN++ [82]	2020	NAS+Attention	Depth	1.7±1.5	2.0±1.2	1.8±0.7
**4**	Massy_HNU [155]	2017	LBP	Texture (HSV+YCbCr)	35.8±35.3	8.3±4.1	22.1±17.6
**4**	GRADIANT [155]	2017	LBP	Texture (HSV/dynamic)	5.0±4.5	15.0±7.1	10.0±5.0
**4**	Auxiliary [33]	2018	CNN+LSTM	Depth+rPPG	9.3±5.6	10.4±6.0	9.5±6.0
**4**	FaceDs [69]	2018	CNN	Texture (Quality/static)	1.2±6.3	6.1±5.1	5.6±5.7
**4**	STASN [79]	2019	CNN+Attention	Texture (dynamic)	6.7±10.6	8.3±8.4	7.5±4.7
**4**	FAS_TD [81]	2019	CNN+LSTM	Depth	14.2±8.7	4.2±3.8	9.2±3.4
**4**	DeepPixBis [70]	2019	DenseNet [131]	Texture (static)	36.7±29.7	13.3±14.1	25.0±12.7
**4**	CDCN [82]	2020	CNN	Depth	4.6±4.6	9.2±8.0	6.9±2.9
**4**	CDCN++ [82]	2020	NAS+Attention	Depth	4.2±3.4	5.8±4.9	5.0±2.9

**Table 6 jimaging-06-00139-t006:** Evaluation of various facial PAD methods on SiW.

Protocol	Method	Year	Feature	Cues	APCER (%)	BPCER (%)	ACER (%)
**1**	Auxiliary [33]	2018	CNN+LSTM	Depth+rPPG	3.58	3.58	3.58
**1**	STASN [79]	2019	CNN+Attention	Texture (dynamic)	-	-	1.0
**1**	FAS_TD [81]	2019	CNN+LSTM	Depth	0.96	0.50	0.73
**1**	CDCN [82]	2020	CNN	Depth	0.07	0.17	0.12
**1**	CDCN++ [82]	2020	NAS+Attention	Depth	0.07	0.17	0.12
**2**	Auxiliary [33]	2018	CNN+LSTM	Depth+rPPG	0.57±0.69	0.57±0.69	0.57±0.69
**2**	STASN [79]	2019	CNN+Attention	Texture (dynamic)	-	-	0.28±0.05
**2**	FAS_TD [81]	2019	CNN+LSTM	Depth	0.08±0.14	0.21±0.14	0.14±0.14
**2**	CDCN [82]	2020	CNN	Depth	0.00±0.00	0.13±0.09	0.06±0.04
**2**	CDCN++ [82]	2020	NAS+Attention	Depth	0.00±0.00	0.09±0.10	0.04±0.05
**3**	Auxiliary [33]	2018	CNN+LSTM	Depth+rPPG	8.31±3.81	8.31±3.80	8.3±3.81
**3**	STASN [79]	2019	CNN+Attention	Texture (dynamic)	-	-	12.10±1.50
**3**	FAS_TD [81]	2019	CNN+LSTM	Depth	3.10±0.81	3.09±0.81	3.10±0.81
**3**	CDCN [82]	2020	CNN	Depth	1.67±0.11	1.76±0.12	1.71±0.11
**3**	CDCN++ [82]	2020	NAS+Attention	Depth	1.97±0.33	1.77±0.10	1.90±0.15

**Table 7 jimaging-06-00139-t007:** Cross-database testing between CASIA-FASD and REPLAY-ATTACK: the reported evaluation metric is Half Total Error Rate (HTER) (%).

Method	Year	Feature	Cues	Train	Test	Train	Test
				CASIA- FASD	REPLAY- ATTACK	REPLAY- ATTACK	CASIA- FASD
LBP [37] ^a^	2012	LBP	Texture (static)	55.9		57.6	
Correlation 19 [189]	2013	MLP	Motion	50.2		47.9	
LBP-TOP [73]	2013	LBP	Texture (dynamic)	49.7		60.6	
Motion Mag [74]	2013	HOOF	Texture+Motion	50.1		47.0	
Yang et al. [34]	2014	CNN	Texture (static)	48.5		45.5	
Spectral cubes [77]	2015	Fourier Spectrum +codebook	Texture (dynamic )	34.4		50.0	
Color texture [35]	2015	LBP	Texture (HSV/static)	47.0		39.6	
Color LBP [60]	2016	LBP	Texture (HSV/static)	30.3		37.7	
Auxiliary [33]	2018	CNN+LSTM	Depth+rPPG	27.6		28.4	
FaceDs [69]	2018	CNN	Texture (Quality/static)	28.5		41.1	
STASN [79]	2019	CNN+Attention	Texture (dynamic)	31.5		30.9	
FAS_TD [81]	2019	CNN+LSTM	Depth	17.5		24.0	
CDCN [82]	2020	CNN	Depth	15.5		32.6	
CDCN++ [82]	2020	NAS+Attention	Depth	6.5		29.8	

^a^ Results taken from [73].

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
