# Peer review of "A Survey on Anti-Spoofing Methods for Facial Recognition with RGB Cameras of Generic Consumer Devices"

_2313-433X, 2020, doi:10.3390/jimaging6120139_

Round 1
Reviewer 1 Report
The paper provides a review of methods for presentation attack detection (PAD) that rely on features from RGB cameras. It discusses different types of algorithms, ranging from behavioral-based over hand-crafted feature-based to modern deep learning-based methods. Additionally, existing PAD datasets are discussed, and results provided in the research papers are enlisted.
Generally, the paper -- though quite lengthy -- contains valuable information and I would recommend publication, after the following issues have been solved:
1. While there exists a relatively novel review [Bhattacharjee2019] on face PAD, this review is neither cited nor discussed by the authors. A good addition would be the discussion of a recent comparison of PAD methods [Komulainen2019].
[Bhattacharjee2019] Bhattacharjee S., Mohammadi A., Anjos A., Marcel S.: "Recent Advances in Face Presentation Attack Detection." Handbook of Biometric Anti-Spoofing. Advances in Computer Vision and Pattern Recognition. Springer, 2019
[Komulainen2019] Komulainen J., Boulkenafet Z., Akhtar Z.: "Review of Face Presentation Attack Detection Competitions." Handbook of Biometric Anti-Spoofing. Advances in Computer Vision and Pattern Recognition. Springer, 2019
2. In line 49 (as well as in the abstract and later in section 5), the authors claim to perform an experimental comparison of the methods. However, no experiments are run by the authors themselves, so they should rephrase their claim: they list results reported in the reviewed papers.
3. In many places in the paper, the authors mention to review the most recent papers on PAD. However, especially the first papers that they review in section 2.2.1 are almost 20 years old now and, therewith, not recent. Again, the authors should rephrase their claims.
4. At the end of section 2.2, the authors should talk about one important issue with liveness-based methods, which is the duration of the recording required to assess liveness. A similar discussion was performed in line 668, but it should be repeated in section 2.2.
5. In line 129, the authors mention that 3D and infrared sensors are only used in expensive capturing devices, but this is not true. Every iphone and, to my knowledge, also every Samsung and Huawei smartphone includes these or similar types of sensors.
6. The first work listed in section 2.3.1 is outdated and does not work in normal conditions. The authors should consider to remove the discussion of that method.
7. There are several minor points that should be taken into consideration:
- Figure 1 lists unknown attacks only for impersonation attacks, but also obfuscation attacks can be unknown.
- Line 391 starts with "as mentioned in the previous section", but talks about something that has not be mentioned in the previous section.
- In line 818, the authors state that ResNets include bypass connections, but ResNets only include residual connections, which are different than bypasses.
- In line 1256, the authors list true positives, true negatives, false positives and false negatives. They should also define, what these mean in the context of PAD, and how they are computed (i.e., based on a certain threshold).
- Table 3 contains one algorithm that has a 0 HTER, which is most probably wrong.
Finally, there are a few spelling and grammar mistakes that the authors should fix, such as:
- line 310: a [...] Machines -> a [...] Machine
- several places: a SVM -> an SVM
- line 523: LD) -> LDA
- line 1157+: MSU-RAFS -> UAD
- figure 43: (middle) is used twice.
Author Response
Thank you very much for your comments. Please check the attachment.

Reviewer 2 Report
This paper surveyed over 50 papers, and provided a summarized overview of the available public databases and an extensive comparison of the results reported in the PAD reviewed papers.
This paper can help the people, who have interest in spoofing presentation attack, to shorten their time on existing algorithms and methods. Moreover, its comparison can inspire and give some new ideas to researchers for their further work.
There existed some problems which need to be revised :
- Row 761 “betwen” has been spelled wrong. It should be “between”.
- Row 917 “previsously” has been spelled wrong. It should be “previously”.
- Row 1356 “resarchers” has been spelled wrong. It should be “researchers”.
- Table 2 Replay-attack’s column of Biometric system acquisition device’s content is not clear.
Author Response
We would like to thank the reviewer for his/her valuable comments.
Taking all the comments into consideration, we have revised the paper as follows. The revised texts in the revised manuscript have been emphasized using magenta color.
- Row 761 “betwen” has been spelled wrong. It should be “between”.
â–¸ As the reviewer suggested, we have corrected the “betwen” as “between” in Line 761 in Page 23 of Section 2.4.2
- Row 917 “previsously” has been spelled wrong. It should be “previously”.
â–¸ We have corrected the “previsously” as “previously” in Line 917 in Page 28 of Section 2.6.2, as the reviewer suggested.
- Row 1356 “resarchers” has been spelled wrong. It should be “researchers”.
â–¸ Thanks again for the reviewer’s careful review. We have corrected “resarchers” as “researchers” in Line 1356 in Page 42 of Section 4.3.2.
- Table 2 Replay-attack’s column of Biometric system acquisition device’s content is not clear.
â–¸ The formatting problem noted by reviewer 3 was due to transforming our LaTeX files into Word format for the second round of reviews. For this third round, we transformed back our files into LaTeX. So, the format of Table 2 in of Section 3.3, Page 33 is now correct. We checked it carefully, and in particular the content of Replay-attack’s column of Biometric system acquisition clearer.
Sincerely yours,
Zuheng MING, Muriel VISANI, Muhammad Muzzamil LUQMAN and Jean-Christophe BURIE
Round 2
Reviewer 1 Report
All my previous points have been addressed satisfactorily by the authors.
Author Response
We would like to thank the reviewer for his/her valuable comments.
Sincerely yours,
Zuheng MING, Muriel VISANI, Muhammad Muzzamil LUQMAN and Jean-Christophe BURIE